# CAN YOU HEAR ME NOW? A BENCHMARK FOR LONG-RANGE GRAPH PROPAGATION

**Luca Miglior**[*]  **Matteo Tolloso**[*]  **Alessio Gravina**[*]  **Davide Bacciu**
Department of Computer Science, University of Pisa, Italy

## ABSTRACT

Effectively capturing long-range interactions remains a fundamental yet unresolved challenge in graph neural network (GNN) research, critical for applications across diverse fields of science. To systematically address this, we introduce ECHO (Evaluating Communication over long HOps), a novel benchmark specifically designed to rigorously assess the capabilities of GNNs in handling very long-range graph propagation. ECHO includes three synthetic graph tasks, namely single-source shortest paths, node eccentricity, and graph diameter, each constructed over diverse and structurally challenging topologies intentionally designed to introduce significant information bottlenecks. ECHO also includes two real-world datasets, ECHO-Charge and ECHO-Energy, which define chemically grounded benchmarks for predicting atomic partial charges and molecular total energies, respectively, with reference computations obtained at the density functional theory (DFT) level. Both tasks inherently depend on capturing complex long-range molecular interactions. Our extensive benchmarking of popular GNN architectures reveals clear performance gaps, emphasizing the difficulty of true long-range propagation and highlighting design choices capable of overcoming inherent limitations. ECHO thereby sets a new standard for evaluating long-range information propagation, also providing a compelling example for its need in AI for science.

## 1 INTRODUCTION

Graphs are fundamental data structures used extensively to represent complex interconnected systems, ranging from social networks and biological pathways, to communication infrastructures and molecular structures. Graph Neural Networks (GNNs) (Sperduti, 1993; Gori et al., 2005; Scarselli et al., 2008; Micheli, 2009; Bruna et al., 2014; Defferrard et al., 2016) have emerged as a successful methodology within deep learning, whose research community was initially driven by the development of diverse architectures capable of capturing intricate relational patterns inherent to graph-structured data, as well as impactful applications across various domains (Hamilton et al., 2017; Derrow-Pinion et al., 2021; Gravina et al., 2022; Bacciu et al., 2024; Gravina & Bacciu, 2024; Khemani et al., 2024; Miglior et al., 2025; Tolloso et al., 2026).

More recently, the research community has shifted its focus towards understanding and overcoming fundamental limitations of the message-passing paradigm underlying GNNs. This shift has been driven by the observation that effectively propagating information over long distances in graphs remains a significant challenge. Such challenges have been formally linked to phenomena like over-smoothing (Cai & Wang, 2020; Oono & Suzuki, 2020; Rusch et al., 2023), over-squashing (Alon & Yahav, 2021; Di Giovanni et al., 2023), and more generally, vanishing gradients (Arroyo et al., 2025), all of which hinder GNN performance in tasks that require capturing long-range dependencies.

Currently, we are in the stage in which such pioneering theoretical studies need consolidation, while looking into methodological advancements that can surpass or mitigate such shortcomings. A key enabler of this progress is the establishment of solid and challenging benchmarks that can accurately assess and validate long-range propagation capacities. The availability of controlled synthetic benchmarks, should be complemented by the introduction of compelling application-driven

---

[*]Equal contribution.
Correspondence: luca.miglior@phd.unipi.it, matteo.tolloso@phd.unipi.it, alessio.gravina@di.unipi.it

datasets which can clearly demonstrate the practical advantages of addressing long-range propagation issues. Long-range propagation capacities, in this sense, have been noted to be central in key areas of science, such as in biology (Dwivedi et al., 2022; Hariri & Vandergheynst, 2024), biochemistry (Gromiha & Selvaraj, 1999), and climate (Lam et al., 2023).

Existing graph benchmarks have, instead, focused primarily on short- to medium-range tasks (Bojchevski & Günnemann, 2018; Shchur et al., 2018; Wu et al., 2018; Sterling & Irwin, 2015; Wale & Karypis, 2006; Hu et al., 2020a; Dwivedi et al., 2023), often overlooking the unique challenges associated with distant information propagation. More recently, the growing interest in this challenge has motivated the community to develop a few benchmarks specifically designed to evaluate information propagation in GNNs. These include the Long-Range Graph Benchmark (LRGB) (Dwivedi et al., 2022) and the Graph Property Prediction (GPP) dataset (Gravina et al., 2023). While this is a significant step forward compared to earlier benchmarks, it does not fully account for the need to capture the true long-range dependencies present in some real-world applications. This is due to limited size of the graphs, the absence of well-defined conditions on the expected propagation range, and the focus of the benchmarks, which is often more aimed at specific issues of over-smoothing and over-squashing, rather than providing a broader evaluation of long-range propagation capabilities. Moreover, LRGB and GPP tasks are facing a natural performance saturation, as novel methodologies are being developed and optimized on them.

Motivated by this, we introduce `ECHO` (Evaluating Communication over long HOps), a new benchmark designed to assess the capabilities of GNNs to exploit long-range interactions. `ECHO` consists of three synthetic tasks and two real-world chemically grounded tasks. The former are designed to provide a controlled setting to assess propagation capabilities. They comprise the prediction of shortest-path-based graph properties (i.e., node eccentricity, single-source shortest paths, and graph diameter) across diverse graph topologies. These have been defined to increase the difficulty of effective long-range communication, as they present structural bottlenecks to information flow. The main characteristic of these tasks is that GNNs must heavily rely on global information and effectively learn to traverse the entire graph, similarly to classical algorithms like Bellman-Ford (Bellman, 1958). The real-world tasks target the prediction of molecular total energy and the long-range charge redistribution in molecules, which are critical and practically relevant challenges in computational chemistry (Dupradeau et al., 2010), as they underlie many fundamental processes such as chemical reactivity, molecular stability, and intermolecular interactions. Accurate modeling of these effects is essential for drug design, materials science, and biology understanding.

Our contributions can be summarized as follows:

- We introduce `ECHO`, a novel benchmark featuring five new tasks specifically designed to evaluate the ability of GNNs to effectively handle long-range communication in both synthetic and real-world settings. `ECHO` includes three synthetic tasks (collectively referred to as `ECHO-Synth`) with a total of 10,080 graphs, and two real-world tasks (collectively referred to as `ECHO-Chem`) comprising between ≈170,000 and ≈196,000 graphs, where the required propagation ranges from 17 to 40 hops.

- We propose `ECHO-Chem`, a novel benchmark consisting of two tasks (i.e., `ECHO-Charge` and `ECHO-Energy`) designed to capture long-range atomic interactions in molecular graphs. Specifically, `ECHO-Charge` is a dataset for predicting atomic charge distributions, while `ECHO-Energy` focuses on predicting the total energy of a molecule. Both tasks are built on Density Functional Theory (DFT) (Argaman & Makov, 2000) calculations, ensuring quantum-level accuracy. This makes them particularly suitable for evaluating long-range message passing in GNNs, since both charge redistribution and molecular energy depend on subtle, non-local effects. Beyond benchmarking, these datasets also address central challenges in computational chemistry, where modeling long-range interactions remains difficult and computationally expensive, as evidenced by the ≈ **2 months** of parallel DFT computations required to generate our benchmark on the given hardware configuration.

- We present a detailed analysis to demonstrate that the tasks in `ECHO` genuinely capture long-range dependencies, providing a rigorous evaluation of GNNs' ability to propagate information over extended graph distances.

- We conduct extensive experiments to establish strong baselines for each task in `ECHO`, providing a comprehensive reference point for future research on long-range graph propagation.

## 2    ON THE NEED FOR A NEW BENCHMARK

We now elaborate on the need for novel benchmarks specialized on the evaluation of long-range propagation in relation to existing datasets.

The most widely used benchmark for assessing these capabilities is arguably LRGB (Dwivedi et al., 2022). Its introduction in 2022 has certainly marked an important milestone and promoted the development of the field. However, despite initial rapid improvements, performance on LRGB has now plateaued, showing a noticeable deceleration in progress across the last year, as discussed in Appendix B. In addition, recent works (Tönshoff et al., 2023; Bamberger et al., 2025b) question the long-range nature of several LRGB tasks, showing that some are inherently local rather than requiring global information, and that the benchmark itself is highly sensitive to hyperparameter tuning. Other benchmarks propose synthetic tasks on generated structures, including the Tree-Neighborhood (Alon & Yahav, 2021), Graph Property Prediction (Gravina et al., 2023), graph transfer (Di Giovanni et al., 2023; Gravina et al., 2025), GLoRA (Zhou et al., 2025), and Barbell and Clique graphs (Bamberger et al., 2025a). Indeed, most of these tasks are originally designed to address narrow challenges that prevent long-range propagation, such as over-smoothing (Cai & Wang, 2020; Oono & Suzuki, 2020; Rusch et al., 2023) and over-squashing (Alon & Yahav, 2021; Di Giovanni et al., 2023). These phenomena, while related, do not necessarily capture the full spectrum of challenges associated with long-range communication. Moreover, despite being designed to test the ability of GNNs to overcome these limitations, these datasets typically involve small graphs with limited-size diameters. This inherently restricts the propagation radius, creating a significant gap between the benchmark tasks and real-world problems that require much deeper propagation across significantly larger structures.

The limitations above suggest the need for a new benchmark that reflects the challenges and opportunities in long-range GNN research. An effective benchmark should provide tasks that explicitly test a model's ability to traverse extensive graph structures, effectively aggregate global information, and adapt to diverse topological constraints. Moreover, as the field has matured and a wide range of models have been established, ranging from graph transformers (Shi et al., 2021; Rampášek et al., 2022) to multi-hop GNNs (Abu-El-Haija et al., 2019; Gutteridge et al., 2023) and others (Shi et al., 2023), it seems timely to introduce a new benchmark that can accurately assess the long-range propagation skills of these families of models, now that they are well understood and consolidated.

`ECHO` addresses this scenario by a suite of synthetic and real-world tasks with clearly defined long-range propagation needs, providing a clear target for the evaluation of this property. `ECHO` tasks require computing shortest paths between all nodes, long-range charge redistribution, or molecular total energies, with clearly defined propagation ranges between 17 and 40 hops, depending on the specific graph structure. This explicit range ensures that models failing to capture dependencies within this span are underreaching and have poor long-range capabilities.

The `ECHO-Charge` and `ECHO-Energy` molecular tasks have strong value per se, proposing a novel, practical, and high-impact challenge for learning models in computational chemistry (Dupradeau et al., 2010). Previous popular benchmarks in this domain (Sterling & Irwin, 2015; Wale & Karypis, 2006; Hu et al., 2020a; Wu et al., 2018; Dwivedi et al., 2022) focused on the prediction of molecular-level properties, such as solubility or HIV inhibition, which are predominantly short-range tasks. This is evident when they can be reduced to the problem of counting small-dimensional local substructures (i.e., with length smaller than 7) (Bouritsas et al., 2023). Differently, `ECHO-Charge` and `ECHO-Energy` are the first graph benchmarks that target long-range interactions at the atomic level, i.e., the microscopic scale. Both benchmarks are not only inherently long-range but also particularly challenging, as they require accurate modeling of charge distributions, energy stabilization, and the complex interplay of atomic interactions. This makes them computationally expensive to solve with current computational chemistry tools. We provide further details on the computational complexity of the underlying quantum simulations in Appendix C.

Therefore, `ECHO-Charge` and `ECHO-Energy` set a new standard for evaluating long-range graph information propagation, as well as they provide a compelling application of AI for science and chemistry, enabling faster predictions with potential impact on drug/material design or understanding biological functions.

Contemporaneously with our work, Liang et al. (2025) proposed a synthetic benchmark, which we view as a complementary effort to our `ECHO` in addressing the long-range propagation problem. While Liang et al. (2025) focuses on a single synthetic task on large graphs (up to 569k nodes), our

ECHO benchmark proposes five tasks (as discussed before) that provide a controlled setting to assess propagation capabilities, are inherently long-range, and extend beyond current standards. Our goal is to provide a practical, accessible benchmark that balances long-range complexity and usability for the broader community, avoiding digital divide concerns while still reflecting real-world scientific challenges.

## 3 THE ECHO BENCHMARK

In this section, we introduce a suite of datasets designed to rigorously evaluate the long-range information propagation capabilities of GNNs. Our benchmark consists of two complementary components: a set of algorithmically constructed tasks (collectively called ECHO-Synth) and a set chemically grounded real-world datasets (collectively called ECHO-Chem). Detailed dataset statistics are reported in Appendix E.

The synthetic component includes classical graph-theoretic problems (i.e., single-source shortest path, node eccentricity, and graph diameter) posed across diverse graph topologies designed to induce structural bottlenecks and challenge multi-hop message passing. These tasks isolate long-range dependencies and enable controlled analysis of model behavior under varying topological conditions.

The proposed real-world benchmarks target practically relevant and physically grounded tasks in computational chemistry: ECHO-Charge focuses on predicting long-range charge redistribution at the atomic level, while ECHO-Energy addresses the prediction of molecular total energies. Both problems are rooted in electronic structure modeling, reflecting realistic quantum phenomena such as charge transfer and energy stabilization, and build upon prior work in quantum-accurate deep learning models for molecular systems (Ko et al., 2021; Zhang et al., 2022).

### 3.1 THE ECHO-SYNTH DATASET

The algorithmic dataset is designed to benchmark GNNs on tasks that require long-range information propagation across a diverse set of graph topologies. It focuses on three graph property prediction tasks: **Single Source Shortest Path** (sssp), **Node Eccentricity** (ecc), and **Graph Diameter** (diam). Among these, sssp and ecc are node-level tasks requiring the prediction of a scalar value per node, while diam is a graph-level task requiring a single prediction for the entire graph. We refer to this dataset as ECHO-Synth.

These tasks draw inspiration from Corso et al. (2020); Gravina et al. (2023)[1] and were intentionally selected due to their heavy reliance on global information. For example, solving sssp from a given source node requires identifying shortest paths to all other nodes (Dijkstra, 2022), since the information spans the entire graph. Eccentricity builds on this by requiring the longest shortest path from each node, demanding complete graph awareness. Diameter is even more global, involving the longest shortest path between *any* two nodes (Cormen et al., 2009). Classical algorithms like Dijkstra's (Dijkstra, 2022) and Bellman-Ford (Bellman, 1958), which perform complete graph traversal, illustrate the challenge these tasks pose for GNNs, which rely on localized message passing. To prevent models from relying on input features rather than learning structural patterns, each node is assigned a uniformly distributed random scalar feature $r \sim \mathcal{U}(0, 1)$.[2] Additionally, for the sssp task, a binary indicator is included to mark the source node. This ensures that the model can distinguish the source while maintaining uniform input statistics across tasks.

---

[1]While ECHO-Synth draws inspiration from Corso et al. (2020); Gravina et al. (2023), it departs from them in a key aspect. Both Corso et al. (2020) and Gravina et al. (2023) use small graphs that are mostly sampled from distributions that yield highly connected structures with small diameters and limited long-range dependencies. In contrast, ECHO-Synth is explicitly designed to rigorously stress-test long-range capabilities, leveraging larger graphs and employing topologies deliberately designed to introduce bottlenecks, i.e., requiring substantially more propagation steps for accurate predictions. These design choices significantly increase the long-range difficulty of the tasks, making ECHO-Synth a more rigorous benchmark for evaluating long-range propagation.

[2]We opted for random uniform node features rather than zero vectors to introduce stochasticity (which makes the synthetic tasks less trivial), and to provide a unique identifier for the nodes in the tasks and prevent the trivial scenario in which all nodes share identical initial representations, hindering the expressiveness of certain architectures (Sato et al., 2021).

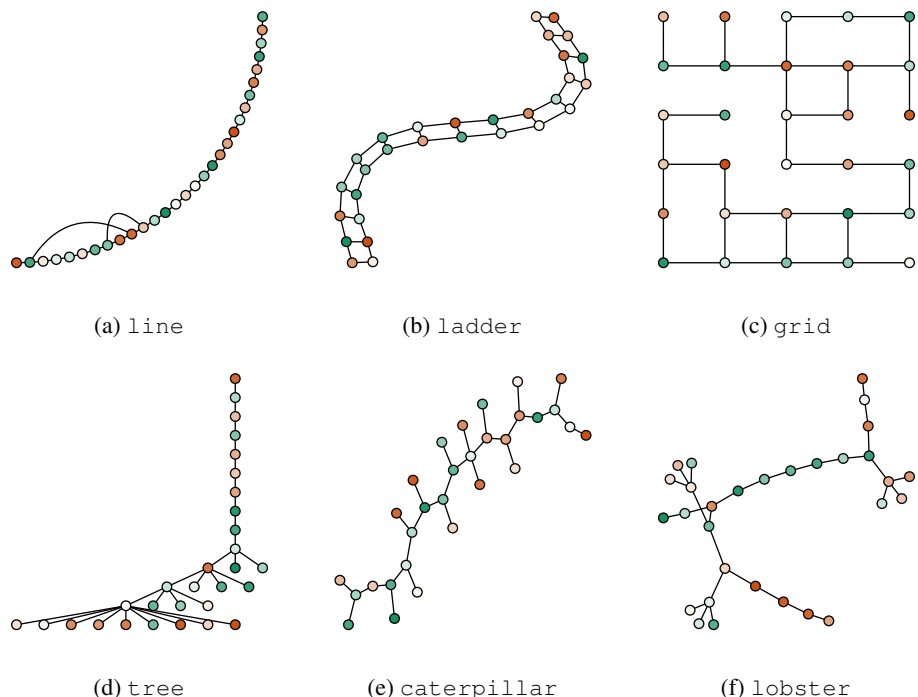

(a) `line`    (b) `ladder`    (c) `grid`

(d) `tree`    (e) `caterpillar`    (f) `lobster`

Figure 1: Visualization of the proposed topologies in the synthetic dataset. All graphs consist of 30 nodes to ease visualization, and different node colors correspond to different random initial features.

**Dataset Construction.** This dataset includes six distinct families of graph topologies i.e., line, ladder, grid, tree, caterpillar, and lobster (see Figure 1), each selected to highlight different structural and propagation characteristics. The `line` graph (Figure 1a) serves as a simple but non-trivial baseline. To introduce non-local interactions, we modify it with stochastic residual connections: each node has a 20% chance of forming an edge to another node 2–6 hops away. Building on this, the `ladder` topology (Figure 1b) consists of two parallel `line` graphs connected by one-to-one cross-links, enabling richer routing possibilities and redundancy in message pathways. The `grid` topology (Figure 1c) represents a 2D lattice structure where edges are independently removed with a 20% probability. This results in irregular neighborhoods and broken spatial symmetries.

To model hierarchical structures, we include `tree`-structured graphs (Figure 1d) generated through preferential attachment. A new node connects to an existing one with probability proportional to $k_i^{\alpha}$, where $k_i$ represents the degree of the $i$-th node (with $\alpha = 3$), leading to the formation of high-degree hubs and reflecting connectivity patterns often seen in natural networks. The `caterpillar` topology (Figure 1e) augments a central linear backbone with peripheral nodes attached randomly along the spine, combining features of chain-like and tree-like graphs to create moderate branching and directional flow. Extending this idea, the `lobster` graph (Figure 1f) adds a third hierarchical layer: nodes in the outermost layer connect only to intermediate nodes, resulting in deeper branching while preserving an overall elongated structure. This configuration is especially useful for testing the limits of multi-hop message passing under structured constraints. Beyond their long-range dependencies, the complexity of the synthetic tasks is further increased by the presence of **topological bottlenecks**, which pose significant challenges to GNN based on message passing (Gilmer et al., 2017). Bottlenecks emerge in graphs where information flow between distant nodes is constrained to pass through a small subset of intermediary nodes, thereby restricting the bandwidth of information flow. This structural constraint can increse the risk of *over-squashing*, a phenomenon in which exponentially growing information is aggregated into the low-dimensional node representations (Alon & Yahav, 2021). As a result, critical signals may be compressed or lost during propagation, severely limiting the model's capacity to distinguish and preserve meaningful long-range interactions (Topping et al., 2022; Di Giovanni et al., 2023).

Graph families in synthetic dataset are explicitly designed to expose models to such bottlenecks. For example, in the `line` topology information between distant nodes must propagate sequentially through a single path, making each node along the path a critical bottleneck. Similarly, `tree`-structured graphs inherently introduce bottlenecks at branch points and hierarchical layers, where entire subtrees depend on narrow pathways for communication with the rest of the graph. The `caterpillar` and `lobster` graphs further reinforce this pattern by adding additional peripheral layers while maintaining centralized backbones, exacerbating the bottleneck effect in their hierarchical layouts. Even in the more uniform `grid` topology, bottlenecks are implicitly introduced through random edge deletions, which can disrupt regular pathways and force information to traverse suboptimal and congested routes.

**Dataset Split.** To support robust evaluation, we generate graphs with target diameters in the range $d \in [17, 40]$, capturing diverse long-range interaction scenarios. For each of the six graph topologies and each diameter value, we produce 70 unique graphs, yielding a total of $70 \times 24 \times 6 = 10,080$ graphs. To ensure consistent and unbiased evaluation, we partition these graphs into training, validation, and test splits in a stratified manner. Specifically, for each topology and diameter combination, we assign 40 graphs to the training set, 15 to the validation set, and 15 to the test set. This strategy guarantees that all splits share the same distribution over both graph topologies and diameter values, which are uniformly sampled. Consequently, models are evaluated on data that is statistically aligned with the training set, avoiding distributional shifts and ensuring fair comparison across methods.

## 3.2 THE ECHO-CHEM DATASETS

Molecular property prediction is a cornerstone application of GNNs, with common benchmarks involving graph-level prediction tasks such as molecular fingerprint (Duvenaud et al., 2015), solubility, toxicity and various chemical properties (Coley et al., 2017; Hu et al., 2020b). One fundamental task in this domain is the prediction of atomic partial charges, which are continuous, atom-level properties that reflect the electron distribution within a molecule. Accurate charge prediction is essential for modeling molecular interactions, reactivity, and electrostatic behavior. Figure 2 illustrates this task on the 3D molecular graph of caffeine, where each atom is colored according to its predicted partial charge. Complementary to this, another central quantum property of molecular systems is the total energy, which governs stability, chemical reactivity, and conformational preferences. Thus, predicting molecular energies is equally important for chemistry applications.

Traditionally, both atomic charges and molecular energies are computed using quantum mechanical methods, especially Density Functional Theory (DFT) (Argaman & Makov, 2000) or related quantum chemical simulations. While these methods provide high accuracy, their computational cost, arising from solving complex equations, limits their scalability to large molecular datasets or high-throughput tasks. Specifically, high-accuracy simulations require several minutes to process a single molecule. We report a quantitative description of DFT simulation efficiency in Appendix C.

Figure 2: The 3D molecular graph of *caffeine* annotated with atomic partial charges. Blue indicates regions of negative partial charge, while red corresponds to positive charge accumulation.

A significant challenge for Machine Learning (ML) methods addressing these prediction tasks is effectively capturing long-range dependencies across molecular graphs. Specifically, here we will refer to "long-range" in the graph space (e.g., nodes separated by many hops), rather than purely spatial distance. The three-dimensional configuration of molecules greatly intensifies this task complexity, as distant atoms in the graph topology can still exert significant influence on electronic properties and total energy (Jensen, 2017; Ko et al., 2021; Shaidu et al., 2024). Specifically, the total molecular energy is computed considering several quantum-mechanical long-range interactions (Jensen, 2017), and, similarly, the partial atomic charges are influenced by non-local electronic effects (Ko et al., 2021; Shaidu et al., 2024).

Such non-trivial, long-range interdependencies become increasingly challenging to model accurately as molecular graph diameter grows. To systematically address this challenge, we introduce **ECHO-Charge** and **ECHO-Energy**, with the specific aim to stress long-range dependencies in

real-world scenarios. `ECHO-Charge` is formulated as a node-level regression problem, where the model must predict the partial charge of each atom in a molecular graph, while `ECHO-Energy` is formulated as a graph-level regression problem, requiring the prediction of the total molecular energy. We note that the total energy cannot be computed as the sum of per-atom energies, since it consists of several quantum-mechanical long-range interactions, e.g., electron–nuclear, electron–electron, and nuclear–nuclear contributions at the chosen level of theory (Jensen, 2017). We collectively refer to `ECHO-Charge` and `ECHO-Energy` as `ECHO-Chem`.

Beyond serving as rigorous benchmarks for GNN architectures, these datasets have strong potential for practical impact in ML applications for science and chemistry. Capturing these sophisticated long-range interactions can significantly improve efficiency of predicting atomic partial charges and molecular energies, while also serving as accurate and computationally inexpensive initialization for subsequent quantum mechanical simulations. Such improvements could substantially accelerate computational chemistry workflows, facilitating rapid exploration of the large molecular space.

**Dataset Construction.** Comprising $\approx$170,000 (`ECHO-Charge`) and $\approx$196,000 (`ECHO-Energy`) molecular graphs selected from the ChEMBL database (Zdrazil et al., 2024), our benchmarks include molecules with graph diameters between 17 and 40, where the interplay between the molecule size and the task ensures the need to work with significant long-range dependencies that thoroughly stress model capabilities. In both `ECHO-Charge` and `ECHO-Energy`, each graph represents a single molecule (see Figure 2), and each node (i.e., atom) is labeled with the atomic number, essential for chemical identity, and spatial distance from the center of mass of the molecule, to provide geometrical context. Edges correspond to chemical bonds, and are labeled with bond type (single, double, triple, or aromatic) and bond length. Notably, this encoding of spatial information is invariant under the action of the E(3) group, meaning that relative geometric features such as distances remain invariant under global 3D rotations, reflections, and translations of the molecular structure. This ensures that the spatial representation respects the underlying symmetries of molecular physics, essential for learning physically consistent models.

To generate the datasets, we employed a two-step approach. Firstly, the generation process began with molecular 3D structure generation starting from ChEMBL SMILES (Weininger, 1988) strings for all molecules satisfying the given diameter constraint. In order to generate molecular conformations we opted for coordinate optimization using the Generalized Amber Force Field (GAFF) (Grimme et al., 2010), a well-established force field specifically designed for optimizing a wide variety of organic and medically relevant compounds. These optimized structures served as initialization for the subsequent quantum chemical calculations to determine accurate structures, partial charges, and molecular energies. Specifically, we employed Density Functional Theory (DFT) to match the required chemical accuracy required for reliable molecular property annotation. All computations were run with the ORCA package for quantum chemistry (Neese, 2022; Neese et al., 2020; Neese, 2023). A detailed description of the quantum simulations is provided in Appendix C, along with information about the computing platform in Appendix D.

**Dataset Split.** To evaluate model performance under consistent and reproducible conditions, we employed a random uniform sampling strategy to split the original datasets. This approach ensures a balanced distribution of molecular structures, charge ranges, and energy levels across the training, validation, and test sets, therefore minimizing potential sampling bias. For `ECHO-Charge`, we adopt an 80/10/10 split for training, validation, and testing, while for `ECHO-Energy` we use a 90/5/5 split.

## 4 EXPERIMENTS

**Baselines.** We consider a diverse set of GNNs baselines that capture core directions in the development of graph neural architectures, spanning from classical GNNs to more recent approaches that demonstrate strong empirical performance in capturing long-range dependencies. As classical baseline models, we include GCN (Kipf & Welling, 2017), GIN (Xu et al., 2019), GINE[3] (Hu et al., 2020b) and GCNII (Chen et al., 2020), which represent standard message-passing frameworks with strong theoretical grounding. We also consider a multi-hop GNN, i.e., DRew (Gutteridge et al., 2023), which adaptively rewire the graph to facilitate more effective propagation across distant nodes. We

---

[3]We added GINE as a baseline to `ECHO-Charge` and `ECHO-Energy` benchmarks to overcome the limitations of GIN to process edge attributes.

evaluate GPS (Rampášek et al., 2022) and GRIT (Ma et al., 2023b), two effective graph transformer that enables long-range propagation via attention mechanism between any pairs of nodes. Finally, we explore the performance of a family of GNNs that draw on principles from dynamical systems theory, namely differential-equation inspired GNNs (DE-GNNs). This includes GraphCON (Rusch et al., 2022), which is designed to address the over-smoothing issue, as well as models explicitly designed to perform long-range propagation, whose architectures are based on non-dissipative or port-Hamiltoninan dynamics, such as A-DGN (Gravina et al., 2023), SWAN (Gravina et al., 2025), and PH-DGN (Heilig et al., 2025).

**Model Architecture and hyperparameter selection.** All models share a unified backbone design to enable a fair comparison. In particular, each model is composed of a linear embedding layer, a stack of GNN layers, and a task-specific readout module. For node-level tasks, the readout is a two-layer MLP applied directly to the node representations. For graph-level tasks, node representations are first aggregated using the mean, max, and sum operations, concatenated, and then processed by a two-layer MLP. This standardization ensures that differences in performance are attributable to the core propagation mechanisms rather than auxiliary architectural choices.

Training follows a consistent protocol across all models. We minimize the base-10 logarithm of the Mean Squared Error loss (MSE), $\log_{10}(\text{MSE}(y_{\text{true}} - y_{\text{target}}))$, since the predicted values can be very small in magnitude and this scale-sensitive loss emphasizes small differences. We use the Adam (Kingma & Ba, 2015) optimizer and adopt Early Stopping based on validation loss with a patience of 50 epochs. The maximum number of training epochs is set to 1000. This procedure ensures convergence while preventing overfitting, and serves as a reference setup to facilitate reproducibility of our results. In order to ensure a fair and robust comparison across all methods and datasets, we employ an extensive hyperparameter optimization protocol. Specifically, for each model-dataset pair, we perform a Bayesian Optimization based on a Gaussian Process prior (Snoek et al., 2012) in the chosen hyperparameter space, spanning 100 trials to explore the respective search space efficiently. We report the complete set of explored hyperparameters for each model, as well as with the selected hyperparameters, in Appendix F. Finally, the best configuration found is validated through four independent training runs, each initialized with a different random seed.

**Results on `ECHO-Synth` dataset.**

We report results on the synthetic benchmarks in Table 1. All the values are reported using the Mean Absolute Error (MAE). Additional metrics are reported in the Appendix H, Table 18. We start observing that models employing global attention mechanisms significantly outperform traditional message-passing frameworks. Specifically, GRIT demonstrates a superior performance on the `sssp` task, achieving a remarkably low MAE of 0.121. In line with literature findings (Dwivedi et al., 2022), this result suggests that incorporating

Table 1: Test MAE (mean with standard deviation as subscript) for each model across the three synthetic tasks: `diam`, `ecc`, and `sssp`. Lower is better. Values are color-coded by performance, with darker green indicating lower error.

| Model | diam ↓ | ecc ↓ | sssp ↓ |
|---|---|---|---|
| A-DGN | $1.151_{\pm 0.038}$ | $4.981_{\pm 0.037}$ | $1.176_{\pm 0.140}$ |
| DRew | $1.243_{\pm 0.047}$ | $\mathbf{4.651}_{\pm 0.020}$ | $1.279_{\pm 0.011}$ |
| GraphCON | $2.969_{\pm 0.189}$ | $5.474_{\pm 0.001}$ | $5.734_{\pm 0.011}$ |
| GCN | $3.832_{\pm 0.262}$ | $5.233_{\pm 0.034}$ | $2.102_{\pm 0.094}$ |
| GCNII | $2.005_{\pm 0.093}$ | $5.241_{\pm 0.030}$ | $2.128_{\pm 0.429}$ |
| GIN | $1.630_{\pm 0.161}$ | $4.869_{\pm 0.092}$ | $2.234_{\pm 0.271}$ |
| GPS | $2.160_{\pm 0.098}$ | $4.758_{\pm 0.021}$ | $0.472_{\pm 0.050}$ |
| GRIT | $\mathbf{1.014}_{\pm 0.046}$ | $5.091_{\pm 0.158}$ | $\mathbf{0.121}_{\pm 0.013}$ |
| PH-DGN | $1.627_{\pm 0.398}$ | $5.068_{\pm 0.126}$ | $1.323_{\pm 0.485}$ |
| SWAN | $1.121_{\pm 0.070}$ | $4.840_{\pm 0.045}$ | $0.896_{\pm 0.232}$ |

transformer-like global attention substantially mitigates inherent limitations in localized message-passing, which are pronounced in classic architectures such as GCN and GIN. This is further supported by the analysis in Appendix J, which shows that the highest attention scores are often assigned to node pairs that are not directly connected and often far apart in the graph. Interestingly, differential-equation-inspired architectures, particularly those employing non-dissipative or port-Hamiltonian formulations (SWAN, A-DGN, and PH-DGN), consistently perform well across tasks, with similar performance metrics. GRIT achieves the lowest MAE on the `diam` task (1.014), closely followed by SWAN and A-DGN. This highlights the benefit of incorporating attention or non-dissipative dynamics to improve long-range information propagation. Moreover, DRew, reveals its effectiveness in the `ecc` task, attaining the lowest MAE (4.651). This success emphasizes the advantage of multi-hop information propagation, thus effectively addressing topological bottlenecks critical for accurately

capturing node eccentricities. Differently, GraphCON does not inherently outperform traditional methods, and show notably weaker performance relative to other models of the same architectural family (e.g., A-DGN and SWAN). Thus, mere message-passing dynamics that only mitigate the over-smoothing issue does not ensure superior performance in long-range tasks.

Finally, traditional message-passing models like GCN demonstrate consistent limitations across all benchmarks, indicative of fundamental constraints in purely localized message-passing architectures when facing extensive long-range dependencies as required in our `ECHO-Synth` benchmark suite. This limitation is most evident in the `diam` task, where GCN records the highest MAE (3.832), underscoring its inadequate capacity for global information aggregation.

**Results on `ECHO-Chem` tasks.**

We detail the performance of all evaluated models on the atomic partial charge and energy prediction task in Table 2. Additional metrics are reported in the Appendix H, Tables 19 and 20. As anticipated, architectures capable of handling long-range dependencies demonstrate a clear advantage on both benchmarks, given the nature of the task which requires precise modeling of subtle interatomic interactions spread across the molecular graph.

Notably, GPS achieves the best performance on `ECHO-Energy` and it is competitive on `ECHO-Charge`, confirming the utility of global attention mechanisms in capturing distant influences that modulate quantum chemical properties, albeit at the cost of increased computational complexity (as shown in Appendix I).

Table 2: Test MAE (mean with standard deviation as subscript) performance across models on the `ECHO-Energy` and `ECHO-Charge` tasks. Lower values are better. Cells are color-coded by performance, with darker green indicating lower error (independently normalized per column).

| Model | ECHO-Energy $\downarrow$ | ECHO-Charge $\downarrow$ $(\times 10^{-3})$ |
|---|---|---|
| A-DGN | $12.486_{\pm 1.621}$ | $6.543_{\pm 0.146}$ |
| DRew | $11.325_{\pm 2.394}$ | $9.086_{\pm 0.473}$ |
| GCN | $28.112_{\pm 1.239}$ | $8.421_{\pm 0.512}$ |
| GCNII | $13.235_{\pm 2.630}$ | $8.829_{\pm 0.021}$ |
| GIN | $47.851_{\pm 10.154}$ | $10.784_{\pm 0.059}$ |
| GINE | $23.558_{\pm 7.568}$ | $7.176_{\pm 0.371}$ |
| GPS | $\mathbf{5.257}_{\pm 0.842}$ | $6.182_{\pm 0.219}$ |
| GRIT | $25.508_{\pm 2.507}$ | $7.134_{\pm 6.090}$ |
| GraphCON | $14.295_{\pm 0.807}$ | $19.629_{\pm 0.195}$ |
| PH-DGN | $16.080_{\pm 1.123}$ | $7.915_{\pm 0.269}$ |
| SWAN | $12.629_{\pm 1.157}$ | $\mathbf{6.109}_{\pm 0.103}$ |

Models like A-DGN and SWAN also produce competitive performance, appearing consistently among the top performers, with SWAN emerging as the best model in `ECHO-Charge`. Their success suggests that imposing non-dissipative priors not only improves the propagation dynamics but also guides the model toward chemically plausible solutions. Interestingly, while DRew outperforms classical GNNs, especially in the `ECHO-Energy` taks, it performs comparatively worse than DE-GNN and transformer-based models.

Traditional message-passing networks, particularly GCN and GIN, again lag behind. These results again confirm the hypothesis that localized aggregation, without mechanisms to improve propagation effectiveness or integrate distant node information, is inadequate for atomic-level charge modeling. The `ECHO-Chem` benchmarks thus clearly illustrate the necessity for architectures that either incorporate global attention or embed non-dissipative dynamics to effectively tackle the intricate and non-local dependencies inherent in these molecular tasks.

We provide a visual depiction of charge prediction accuracy on a non-trivial molecule from the test set in Figure 3: the figure compares prediction errors between GPS and GCN. GPS shows consistently lower errors, especially at peripheral atoms, highlighting its ability to capture long-range interactions, while GCN accumulates errors at structurally distant or chemically sensitive sites. This comparison illustrates the benefit of long-range information propagation on complex graphs and chemical structures.

Lastly, we note that although the energies and partial charges errors are small in absolute magnitude across baselines, even subtle deviations (as stated in Dupradeau et al. (2010)), on the order of $10^{-4}\,e$ to $10^{-6}\,e$, can lead to significant downstream effects in molecular modeling and reproducibility of results. Therefore, predictive models must target this level of granularity to produce chemically meaningful output.

**Additional Experiments and Analysis.**
We provide further results and a detailed evaluation of baseline performance in Appendix G. Specifically, we investigate how the radius of the explored neighborhood affects each method (Appendix G.1) and how model performance varies across graphs with different diameters (Appendix G.2). Our results consistently indicate that deeper networks outperform shallower ones, confirming the long-range nature of these benchmarks. Moreover, in Appendix G.3 we examine the impact of the readout depth and show that the final performance appears to be independent of this design choice. In Appendix G.4, we further analyze model performance across different graph topologies in ECHO-Synth. The results indicate that, although absolute per-

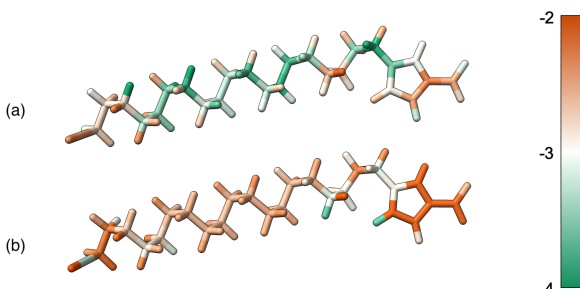

Figure 3: Visualization of prediction errors for the ECHO-Charge task using two different GNN architectures: (a) GPS and (b) GCN. The coloring represents the logarithm of the absolute prediction error, $\log(|y_{\text{true}} - y_{\text{pred}}|)$. Lower values (in green) indicate better prediction accuracy, while higher values (in orange) correspond to larger errors.

formance varies slightly with the underlying graph structure, the relative ranking of the models remains consistent, reinforcing the robustness of our findings. In Appendix J, we also visualize GPS's attention patterns, highlighting the importance of connecting distant nodes to facilitate information flow and improve performance. Together, these analyses reinforce the importance of long-range information propagation in the ECHO benchmark. Finally, Appendix I reports runtime measurements, illustrating a key trade-off between accuracy and efficiency: transformer-based models like GPS achieve strong performance but are computationally demanding, whereas models such as A-DGN provide a more balanced alternative.

## 5 CONCLUSION

In this paper we propose ECHO, a new benchmark for evaluating long-range information propagation in GNNs. Our benchmark includes two main components, ECHO-Synth and a set of chemically grounded real-world datasets (collectively referred to as ECHO-Chem), that target long-range communication in both synthetic and real-world settings. The synthetic tasks are designed to predict algorithmic and long-range-by-design graph properties, while the real-world tasks focus on predicting atomic charge distributions and molecular total energies, both of which critically depend on long-range quantum interactions. We provided a detailed analysis to demonstrate that the tasks in ECHO genuinely capture long-range dependencies, and we established strong baselines for each task to provide a comprehensive reference point for future research. Our results highlight the limitations of current GNN architectures when faced with long-range propagation challenges, and we believe that ECHO will serve as a critical step toward building more robust, scalable, and generalizable GNNs capable of handling the full spectrum of graph-based learning tasks, posing a challenge to the community to push the boundaries of GNN design and evaluation. Not only ECHO provides a solid benchmark, but it also leaves ample room for future architectures to improve and advance GNN architectures capable of more effective information propagation.

## ETHICS STATEMENT

The research conducted in this paper conforms in every aspect with the ICLR Code of Ethics. Our study does not involve human subjects, sensitive personal data, or applications with foreseeable harmful consequences. No ethical concerns are anticipated regarding data usage, methodology, or findings.

## REPRODUCIBILITY STATEMENT

We provide all necessary details to reproduce our ECHO benchmark in Section 3 and Appendix C, and describe the setup of each experiment in Section 4 and Appendix F, thus ensuring sufficient

information to replicate our results. We openly release data and code to replicate our results and perform new evaluations at `https://github.com/Graph-ECHO-Benchmark/ECHO`.

ACKNOWLEDGMENTS

This work has been partially supported by EU-EIC EMERGE (Grant No. 101070918) and NextGeneration EU programme PNNR Partenariato Esteso PE00000013 "FAIR" - Future Artificial Intelligence Research" - Spoke 1 "Human-centered AI".

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

## A    LLMs Usage

Large Language Models (LLMs) were used as general-purpose assistive tools to improve the writing quality of this paper. Specifically, we used LLMs to help with grammar correction, rephrasing for clarity, and suggesting some improvements to the overall structure of the text. All LLM-generated text was carefully reviewed and edited by the authors to ensure that it accurately reflects the authors' intentions and scientific content. No LLMs were used to generate scientific content, including but not limited to research direction, hypothesis formulation, experimental design, data analysis, or interpretation of results.

## B    Discussion on LRGB

One of the most widely used benchmarks for assessing the long-range propagation capabilities of GNNs is the Long Range Graph Benchmark (LRGB) (Dwivedi et al., 2022). The benchmark proposes five tasks: two molecular property prediction tasks (Peptides-func and Peptides-struct), one molecular bond prediction task (PCQM-Contact), and two computer vision tasks (PascalVOC-SP and COCO-SP). However, despite initial rapid improvements, performance on LRGB has plateaued. Since its introduction in 2022, there has been a noticeable deceleration in progress. Considering a set of 30 models on the Peptides-func task, we observe a performance improvement of 6.5% in the first year, but only by 1.3% in the second, and no significant gain in the third year (Dwivedi et al., 2022; Rampášek et al., 2022; Tönshoff et al., 2023; Shirzad et al., 2023; Cai et al., 2023; Glickman & Yahav, 2023; He et al., 2023; Gutteridge et al., 2023; Giusti et al., 2023; Ma et al., 2023a; Ngo et al., 2023; Michel et al., 2023; Ma et al., 2023b; Ding et al., 2024; Choi et al., 2024; Geisler et al., 2024; Behrouz & Hashemi, 2024; Wang et al., 2024; Errica et al., 2024; Gravina et al., 2025; Heilig et al., 2025; Eliasof et al., 2025; Hariri et al., 2025; Trenta et al., 2025). A similar trend exists for the other benchmark tasks as well.

Furthermore, a recent analysis on LRGB (Bamberger et al., 2025b), as well as the benchmark's sensitivity to hyperparameter tuning (Tönshoff et al., 2023), raises additional concerns about the long-range nature of its tasks. The analysis reveals that only a subset of tasks genuinely require longer interactions, while the peptides tasks are effectively local. This highlights the need for more focused benchmarks that explicitly and systematically test the long-range propagation capabilities of GNNs.

## C    Chemical Simulation Technical Information

This appendix provides detailed information on the computational pipeline used to derive partial atomic charges in the `ECHO-Charge` dataset and total energies in `ECHO-Energy`. The pipeline comprises three primary stages: (i) 3D structure generation from SMILES, (ii) quantum chemical computation of partial charges, and (iii) geometry optimization.

**3D Structure Generation from SMILES.** Since subsequent charge optimization steps require pre-optimized 3D coordinates, all structures were geometry-optimized prior to simulation using Open Babel (O'Boyle et al., 2011) and its Python interface, Pybel (O'Boyle et al., 2008) Initial molecular geometries were generated from SMILES strings using the General AMBER Force Field (GAFF) (Grimme et al., 2010). GAFF was chosen over alternatives such as MMFF94 (Halgren, 1996) due to its favorable trade-off between accuracy and computational cost, and its strong performance in predicting both energies and geometries. The optimization procedure involved 100 steps of coarse minimization followed by 500 steps of local refinement for each molecule. The SMILES strings were converted into 3D conformers, which were then minimized to yield low-energy structures. These structures were exported in SDF format for subsequent compatibility. The average time required for 3D structure generation per molecule (considering only those satisfying the `ECHO-Charge` and `ECHO-Energy` dataset diameter criteria) was $\mathbf{562 \pm 124}$ ms.

**Quantum Chemical Computations with ORCA.** To compute partial atomic charges and total energies, we employed the ORCA quantum chemistry software suite (version 6.0.1) (Neese, 2022; 2023; Neese et al., 2020). All calculations were performed using the `B3LYP`, a hybrid density functional (DFT) method that mixes Hartree–Fock exchange with Becke's exchange and Lee–Yang–Parr correlation functionals to balance accuracy and efficiency in quantum chemical calculations (Argaman & Makov, 2000).

Table 3: Mean time required for computation of partial charges of a single molecule with different configurations of the ORCA tool. Variance is computed over 30 random molecules from the `ECHO-Charge`/`ECHO-Energy` dataset. *Denotes the chosen basis set for DFT computation.

| Method | Setting | Times (s) |
|---|---|---|
| HF-3c | LooseSCF | $10.4_{\pm 1.3}$ |
| HF-3c | TightSCF | $28.1_{\pm 4.3}$ |
| B3LYP def2-TZVP* (DFT) | LooseSCF | $146.5_{\pm 10.1}$ |
| **B3LYP def2-TZVP* (DFT)** | **TightSCF** | $\mathbf{634.5_{\pm 21.3}}$ |

We provide a summary of required times for computation with both full DFT computations and HF methods in Table 3. Under our configuration, the average runtime for a single quantum chemical calculation was $\mathbf{634.5 \pm 21.3}$ seconds per molecule, requiring $\approx$ **2 months** of computational time on our hardware configuration. Simulations were run exploiting full thread parallelism provided by ORCA.

**Self-Consistent Field (SCF) Convergence Settings.** To further improve the accuracy of the simulations, we employed the `TightSCF` setting in ORCA, which enforces tighter convergence thresholds in the self-consistent field (SCF) procedure, thereby reducing numerical errors in the electronic structure calculation.

**Charge Extraction.** Atomic partial charges were extracted using the Löwdin population analysis method (Szabo & Ostlund, 1989). These charges were used as supervision signals in our dataset generation pipeline.

# D HARDWARE RESOURCES

All quantum chemistry simulations were conducted on a dual-socket Intel Xeon 6780E machine with a total of 288 physical cores (144 cores per socket, 1 thread per core). Each socket is equipped with 108 MiB of L3 cache, for a combined 216 MiB of shared L3 cache, along with 288 MiB of L2 and 27 MiB of L1 (data + instruction) cache across the system. The CPUs support AVX2 and FMA instruction sets, enabling efficient linear algebra operations, which are critical for electronic structure methods.

The machine is configured with two NUMA nodes, each associated with one of the sockets. Each NUMA node has over 500 GiB of local RAM for a total of approximately 1 TiB of RAM. The high memory capacity and bandwidth are critical for quantum chemistry workloads, particularly those using density functional theory (DFT) or correlated wavefunction methods, which require extensive memory for large basis sets and integral evaluations.

The large number of physical cores allowed us to parallelize over both molecular batches and internal basis function evaluations, providing efficient scaling for density functional theory (DFT) and semi-empirical calculations.

For model training and inference, we used a separate compute node equipped with 8 NVIDIA H100 GPUs.

# E ADDITIONAL DATASET INFORMATION

We report in Table 4 the detailed statistics of the proposed datasets. In Table 5 we provide a summary of the input and target features used in the `ECHO-Synth` and `ECHO-Chem` (i.e., `ECHO-Charge` and `ECHO-Energy`) datasets. Figures 4, 5 and 6 report detailed statistics on the structural properties of the graphs in the datasets, including distributions of the number of nodes, number of edges, average node degree, graph diameter, and node eccentricity. Additionally, Figure 7 illustrates the correlation between the number of nodes and the graph diameter, highlighting structural differences between real and synthetic data. These insights support the design choices for model evaluation across diverse graph regimes.

Table 4: Statistics of the proposed dataset.

| Dataset | # Graphs | Avg Nodes | Avg Deg. | Avg Edges | Avg Diam | # Node Feat | # Edge Feat | # Tasks |
|---|---|---|---|---|---|---|---|---|
| ECHO-Synth | 10,080 | $83.69_{\pm 66.24}$ | $2.53_{\pm 1.19}$ | $211.63_{\pm 209.39}$ | $28.50_{\pm 6.92}$ | 2 | None | 3 |
| line | 1,680 | $75.60_{\pm 27.32}$ | $2.37_{\pm 0.10}$ | $90.10_{\pm 33.89}$ | $28.50_{\pm 6.92}$ | 2 | None | 3 |
| ladder | 1,680 | $56.52_{\pm 13.82}$ | $2.92_{\pm 0.02}$ | $82.54_{\pm 20.72}$ | $28.50_{\pm 6.92}$ | 2 | None | 3 |
| grid | 1,680 | $193.10_{\pm 93.10}$ | $2.95_{\pm 0.12}$ | $288.32_{\pm 145.29}$ | $28.50_{\pm 6.92}$ | 2 | None | 3 |
| tree | 1,680 | $60.42_{\pm 17.17}$ | $1.96_{\pm 0.01}$ | $59.42_{\pm 17.17}$ | $28.50_{\pm 6.92}$ | 2 | None | 3 |
| caterpillar | 1,680 | $34.71_{\pm 7.96}$ | $1.94_{\pm 0.02}$ | $33.71_{\pm 7.96}$ | $28.50_{\pm 6.92}$ | 2 | None | 3 |
| lobster | 1,680 | $81.79_{\pm 25.46}$ | $1.97_{\pm 0.01}$ | $80.79_{\pm 25.46}$ | $28.50_{\pm 6.92}$ | 2 | None | 3 |
| ECHO-Chem | | | | | | | | |
| ECHO-Charge | 170,367 | $72.49_{\pm 12.48}$ | $2.09_{\pm 0.04}$ | $151.32_{\pm 25.16}$ | $23.54_{\pm 2.54}$ | 2 | 2 | 1 |
| ECHO-Energy | 196,528 | $73.73_{\pm 13.22}$ | $2.09_{\pm 0.04}$ | $153.84_{\pm 26.58}$ | $23.61_{\pm 2.59}$ | 2 | 2 | 1 |

Table 5: Summary of dataset properties.

| Dataset | Node Features | Edge Features | Target |
|---|---|---|---|
| ECHO-Synth | Random scalar, source indicator for sssp | None | diam, sssp, ecc |
| ECHO-Chem | Atomic number, distance from center of mass | Bond type, bond length | Partial charges, Total energy |

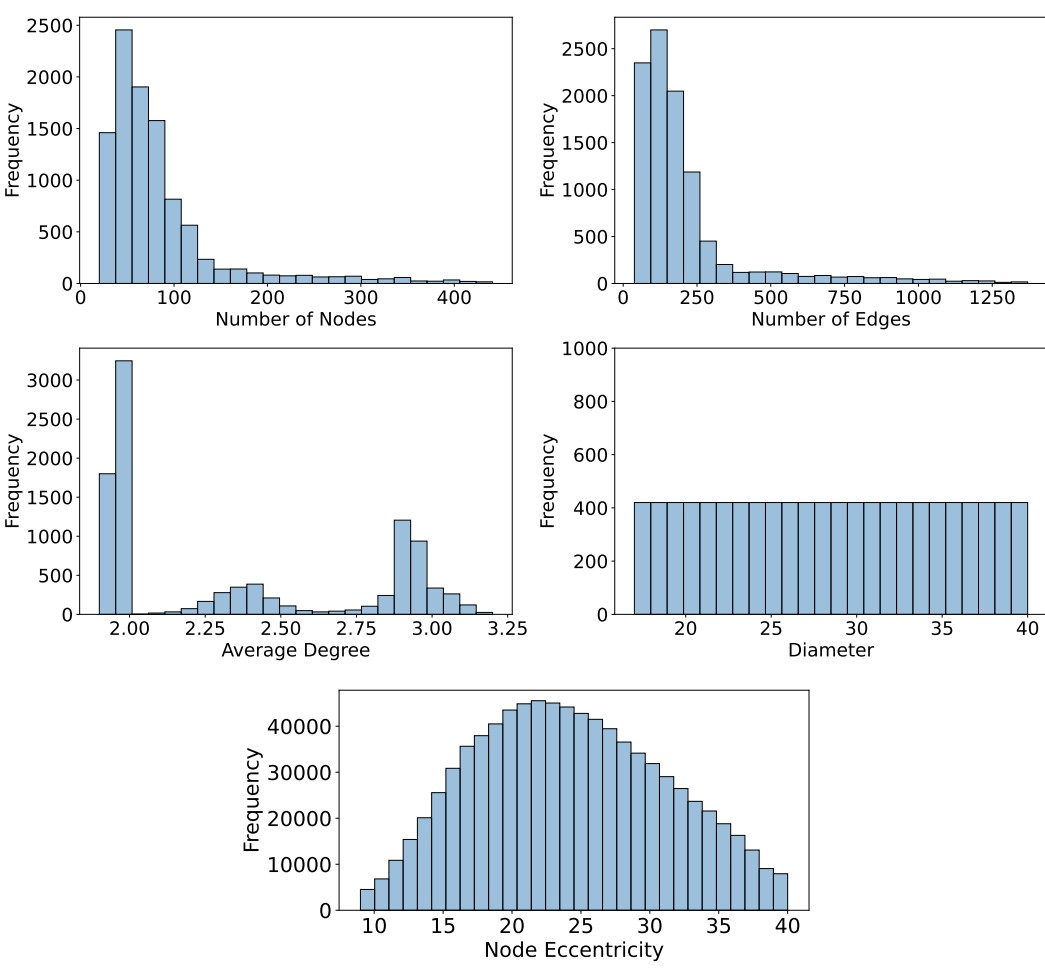

Figure 4: Statistics of the ECHO-Synth dataset, reporting the distributions of the total number of nodes, total number of edges, average degree, diameter, and node eccentricity.

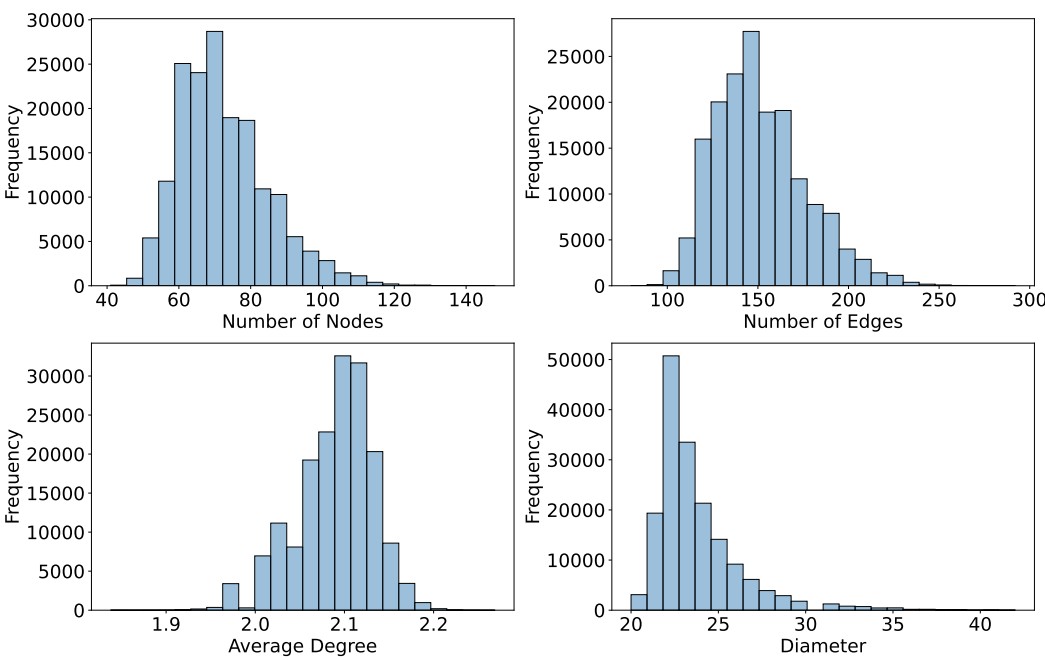

Figure 5: Statistics of the `ECHO-Charge` dataset, reporting the distributions of the total number of nodes, total number of edges, average degree, and diameter.

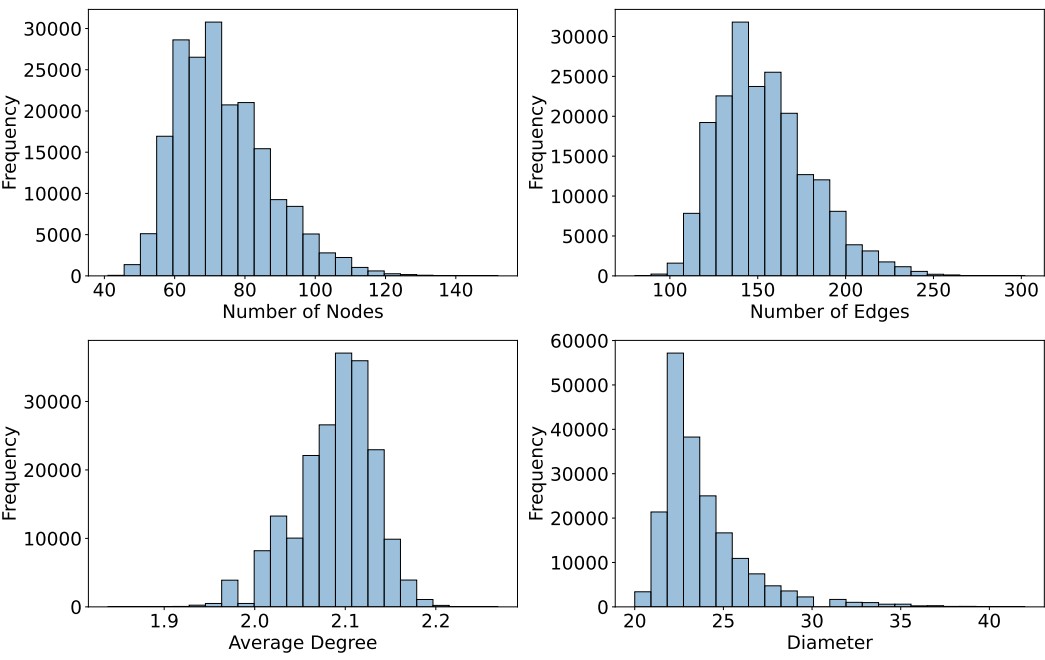

Figure 6: Statistics of the `ECHO-Energy` dataset, reporting the distributions of the total number of nodes, total number of edges, average degree, and diameter.

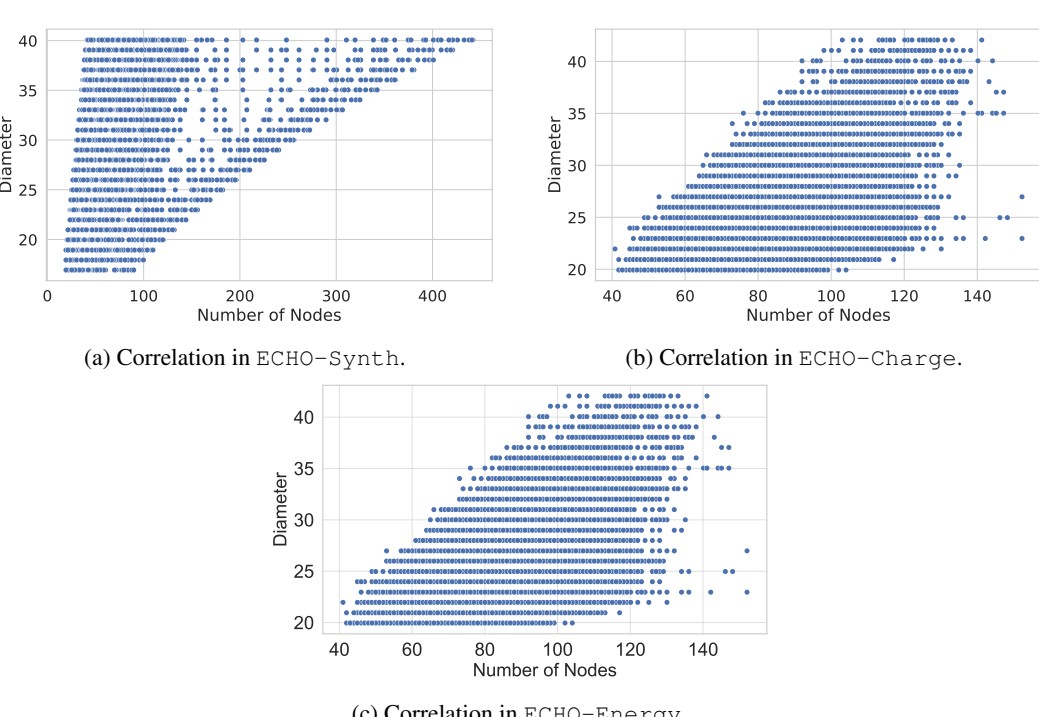

(a) Correlation in `ECHO-Synth`.

(b) Correlation in `ECHO-Charge`.

(c) Correlation in `ECHO-Energy`.

Figure 7: Correlation between the number of nodes and graph diameter in `ECHO-Synth`, `ECHO-Charge` and `ECHO-Energy`.

# F  HYPERPARAMETER SELECTION

Tables 6 to 16 report the hyperparameter search space and the best values selected for each task (`diam`, `ecc`, `sssp`, `ECHO-Charge`, and `ECHO-Energy`) across the different GNN architectures we considered. For details on specific hyperparameters, we refer the reader to the original papers. Each table includes the name of the hyperparameter, its search range or categorical options, and the optimal value obtained for each task, as identified through hyperparameter tuning on the validation set.

Another strong evidence supporting the long-range nature of the `ECHO` benchmark, implicitly comes from our hyperparameter optimization process. Specifically, Bayesian Optimization consistently selected configurations with a large number of GNN layers. This suggests that, even without explicit guidance, the hyperparameter optimization procedure identifies deeper models as necessary to minimize validation error, further reinforcing the notion that the task demands long-range information propagation.

Table 6: Hyperparameters and their best values across tasks for A-DGN.

| Hyperparameter | Search interval | diam | sssp | ecc | ECHO-Charge | ECHO-Energy |
|---|---|---|---|---|---|---|
| Number of layers | $[1 - 40]$ | 27 | 28 | 40 | 34 | 16 |
| Hidden dimension | $[16 - 256]$ | 68 | 65 | 45 | 130 | 216 |
| Learning rate | $[10^{-5} - 10^{-2}]$ | 0.00101 | 0.00229 | 0.00473 | 0.00072 | 0.0085223 |
| Weight decay | $[10^{-8} - 10^{-3}]$ | 0.00098 | 0.00000 | 0.00003 | 0.00001 | 0.00044851 |
| $\epsilon$ | $[0.001 - 0.5]$ | 0.19254 | 0.32934 | 0.10560 | 0.25667 | 0.4215 |
| $\gamma$ | $[0.001 - 0.5]$ | 0.41827 | 0.46803 | 0.21252 | 0.19499 | 0.15304 |
| Graph convolution | $[\text{NaiveAggr}, \text{GCN}]$ | NaiveAggr | NaiveAggr | NaiveAggr | GCN | NaiveAggr |

Table 7: Hyperparameters and their best values across tasks for DRew.

| Hyperparameter | Search interval | diam | sssp | ecc | ECHO-Charge | ECHO-Energy |
|---|---|---|---|---|---|---|
| Number of layers[4] | $[1 - 4]$ | 4 | 4 | 4 | 4 | 3 |
| k-hop | $[1 - 10]$ | 10 | 10 | 10 | 10 | 10 |
| Hidden dimension | $[16 - 256]$ | 249 | 78 | 119 | 232 | 241 |
| Learning rate | $[10^{-5} - 10^{-2}]$ | 0.00037 | 0.00797 | 0.00126 | 0.00036 | 0.00023597 |
| Weight decay | $[10^{-8} - 10^{-3}]$ | 0.00068 | 0.00011 | 0.00003 | 0.0 | 0.00001099 |
| Employ delay | $[\text{True}, \text{False}]$ | False | False | False | True | True |

Table 8: Hyperparameters and their best values across tasks for GCNII.

| Hyperparameter | Search interval | diam | sssp | ecc | ECHO-Charge | ECHO-Energy |
|---|---|---|---|---|---|---|
| Number of layers | $[10 - 40]$ | 32 | 39 | 30 | 37 | 21 |
| Hidden dimension | $[16 - 256]$ | 81 | 40 | 64 | 33 | 103 |
| Learning rate | $[10^{-5} - 10^{-2}]$ | 0.00260 | 0.00032 | 0.00005 | 0.00345 | 0.0047014 |
| Weight decay | $[10^{-8} - 10^{-3}]$ | 0.00000 | 0.00009 | 0.00009 | 0.00002 | 0.000035227 |
| $\alpha$ | $[0.0 - 0.9]$ | 0.70544 | 0.07902 | 0.04742 | 0.17158 | 0.10116 |

---

[4]While the layer search goes up to 4 layers, DRew performs multi-hop aggregation (up to 10 hops per layer), yielding an effective receptive field of $4 \times 10 = 40$ hops, comparable to the ranges explored by the other architectures.

Table 9: Hyperparameters and their best values across tasks for GCN.

| Hyperparameter | Search interval | diam | sssp | ecc | ECHO-Charge | ECHO-Energy |
|---|---|---|---|---|---|---|
| Number of layers | $[1-40]$ | 26 | 40 | 26 | 8 | 9 |
| Hidden dimension | $[16-256]$ | 48 | 42 | 40 | 109 | 160 |
| Learning rate | $[10^{-5}-10^{-2}]$ | 0.00007 | 0.00004 | 0.00023 | 0.00079 | 0.00052181 |
| Weight decay | $[10^{-8}-10^{-3}]$ | 0.00007 | 0.00009 | 0.00002 | 0.00002 | 0.000043613 |

Table 10: Hyperparameters and their best values across tasks for GIN.

| Hyperparameter | Search interval | diam | sssp | ecc | ECHO-Charge | ECHO-Energy |
|---|---|---|---|---|---|---|
| Number of layers | $[10-40]$ | 29 | 34 | 25 | 11 | 19 |
| Hidden dimension | $[16-256]$ | 58 | 170 | 78 | 197 | 90 |
| Learning rate | $[10^{-5}-10^{-2}]$ | 0.00002 | 0.00003 | 0.00006 | 0.00002 | 0.000046134 |
| Weight decay | $[10^{-8}-10^{-3}]$ | 0.00003 | 0.00036 | 0.00091 | 0.00069 | 0.00046213 |

Table 11: Hyperparameters and their best values across tasks for GINE.

| Hyperparameter | Search interval | diam | sssp | ecc | ECHO-Charge | ECHO-Energy |
|---|---|---|---|---|---|---|
| Number of layers | $[1-40]$ | – | – | – | 22 | 31 |
| Hidden dimension | $[16-256]$ | – | – | – | 85 | 33 |
| Learning rate | $[10^{-5}-10^{-2}]$ | – | – | – | 0.00014 | 0.0005028 |
| Weight decay | $[10^{-8}-10^{-3}]$ | – | – | – | 0.00004 | 0.00033338 |

Table 12: Hyperparameters and their best values across tasks for GPS.

| Hyperparameter | Search interval | diam | sssp | ecc | ECHO-Charge | ECHO-Energy |
|---|---|---|---|---|---|---|
| Number of layers | $[1-40]$ | 17 | 26 | 17 | 36 | 26 |
| Hidden dimension | $[16-256]$ | 40 | 56 | 162 | 216 | 192 |
| Learning rate | $[10^{-5}-10^{-2}]$ | 0.00004 | 0.00031 | 0.00034 | 0.00005 | 0.000024067 |
| Weight decay | $[10^{-8}-10^{-3}]$ | 0.00015 | 0.00029 | 0.00007 | 0.00005 | 0.00038179 |
| GNN Backbone | [GCN] | GCN | GCN | GCN | GCN | GCN |
| Number of Backbone Layers | [1] | 1 | 1 | 1 | 1 | 1 |
| Number of attention heads | [2] | 2 | 2 | 2 | 2 | 2 |

Table 13: Hyperparameters and their best values across tasks for GraphCON.

| Hyperparameter | Search interval | diam | sssp | ecc | ECHO-Charge | ECHO-Energy |
|---|---|---|---|---|---|---|
| Number of layers | $[1-40]$ | 37 | 25 | 19 | 35 | 32 |
| Hidden dimension | $[16-256]$ | 63 | 72 | 151 | 144 | 96 |
| Learning rate | $[10^{-5}-10^{-2}]$ | 0.00088 | 0.00013 | 0.00007 | 0.00292 | 0.00032 |
| Weight decay | $[10^{-8}-10^{-3}]$ | 0.00038 | 0.00001 | 0.00001 | 0.00026 | 0.00059 |
| $\epsilon$ | $[0.001-1.0]$ | 0.57880 | 0.95470 | 0.98433 | 0.78163 | 0.82108 |

Table 14: Hyperparameters and their best values across tasks for GRIT.

| Hyperparameter | Search interval | diam | sssp | ecc | ECHO-Charge | ECHO-Energy |
|---|---|---|---|---|---|---|
| Number of layers | $[1-40]$ | 32 | 40 | 32 | 32 | 8 |
| Hidden dimension | $[16-256]$ | 256 | 128 | 128 | 128 | 64 |
| Learning rate | $[10^{-5}-10^{-2}]$ | 0.00082 | 0.00048 | 0.00003 | 0.00034 | 0.00076 |
| Weight decay | $[10^{-8}-10^{-3}]$ | 0.00032 | 0.00047 | 0.00001 | 0.00034 | 0.00094 |
| Attention dropout | $[0-0.5]$ | 0.433 | 0.008 | 0.014 | 0.351 | 0.178 |
| Number of attention heads | [2] | 2 | 2 | 2 | 2 | 2 |

Table 15: Hyperparameters and their best values across tasks PH-DGN.

| Hyperparameter | Search interval | diam | sssp | ecc | ECHO-Charge | ECHO-Energy |
|---|---|---|---|---|---|---|
| Number of layers | $[1-40]$ | 17 | 37 | 21 | 14 | 10 |
| Hidden dimension | $[16-256]$ | 28 | 66 | 120 | 103 | 166 |
| Learning rate | $[10^{-5}-10^{-2}]$ | 0.00150 | 0.00178 | 0.00037 | 0.00033 | 0.0038211 |
| Weight decay | $[10^{-8}-10^{-3}]$ | 0.00054 | 0.00082 | 0.00081 | 0.00063 | 0.00044422 |
| $\epsilon$ | $[0.001-1.0]$ | 0.34977 | 0.16491 | 0.36140 | 0.68993 | 0.40992 |
| $\alpha$ | $[0.01-1.0]$ | 0.47190 | 0.90892 | 0.63323 | 0.87607 | 0.15544 |
| $\beta$ | $[0.01-1.0]$ | 0.70474 | 0.92918 | 0.99675 | 0.91251 | 0.11011 |
| $p$ conv mode | [NaiveAggr, GCN] | GCN | GCN | GCN | GCN | NaiveAggr |
| $q$ conv mode | [NaiveAggr, GCN] | GCN | GCN | GCN | NaiveAggr | NaiveAggr |
| Doubled dimension | [True, False] | False | False | True | True | false |
| Final state | $[p, q, pq]$ | $pq$ | $p$ | $pq$ | $pq$ | $q$ |
| Dampening mode | [param, param+, MLP4ReLU, DGNReLU,] | param+ | DGNReLU | param | param+ | param+ |
| External mode | [MLP4Sin, DGNtanh] | MLP4Sin | MLP4Sin | MLP4Sin | MLP4Sin | MLP4Sin |

Table 16: Hyperparameters and their best values across tasks for SWAN.

| Hyperparameter | Search interval | diam | sssp | ecc | ECHO-Charge | ECHO-Energy |
|---|---|---|---|---|---|---|
| Number of layers | $[1-40]$ | 28 | 40 | 32 | 38 | 25 |
| Hidden dimension | $[16-256]$ | 167 | 97 | 195 | 163 | 217 |
| Learning rate | $[10^{-5}-10^{-2}]$ | 0.00040 | 0.00107 | 0.00086 | 0.00063 | 0.00012157 |
| Weight decay | $[10^{-8}-10^{-3}]$ | 0.00057 | 0.00011 | 0.00010 | 0.00016 | 0.00028686 |
| $\epsilon$ | $[0.001-1.0]$ | 0.54847 | 0.45462 | 0.07451 | 0.38229 | 0.67265 |
| $\gamma$ | $[0.001-0.5]$ | 0.41480 | 0.28342 | 0.45928 | 0.07794 | 0.3156 |
| $\beta$ | $[-1.0-1.0]$ | 0.34233 | -0.20976 | 0.37682 | -0.36245 | -0.67256 |
| Graph convolution | [AntiSymNaiveAggr (ASNA), BoundedGCNConv (BGC), BoundedNaiveAggr (BNA)] | ASNA | BNA | BNA | ASNA | BNA |
| Attention | [True, False] | True | False | False | False | True |

## G    ADDITIONAL EXPERIMENTAL ANALYSIS

In this section, we investigate how the radius of the explored neighborhood influences the performance of each method, as well as how the models perform across graphs with varying diameters. We also analyze how the performance of the models changes across different graph topologies and the influence of the readout depth in `ECHO-Synth`.

### G.1    LAYER-WISE PERFORMANCE ANALYSIS

In Figure 8, we evaluate the impact of the radius of the explored neighborhood (i.e., the number of GNN layers) on test MAE across all tasks. We divide the results into three regimes: shallow ($< 10$ layers), medium (10–17 layers), and deep ($> 17$ layers). Therefore, in the shallow regime, GNNs perform short-range propagation; in the medium regime, they capture medium-range dependencies; and in the deep regime, they are able to model long-range interactions. A consistent pattern emerges across most tasks: deeper networks, especially those tailored for long-range propagation, tend to perform better, thus confirming the long-range nature of the proposed benchmarks. Specifically:

- On the `diam` task (Figure 8a), performance trends are model-dependent. Long-range models such as DRew and A-DGN remain stable or slightly improve, and PH-DGN exhibits a large performance improvement moving from a shallow to a medium depth regime. This task, being graph-level and heavily reliant on global information by design, clearly benefits from increased depth and non-dissipative architectures which are able to perform many message-passing steps across multiple hops.

- For the `ecc` task (Figure 8b), we observe a consistent performance gain with increasing depth across nearly all models. Again, long-range architectures like A-DGN and SWAN, or the multi-hop GNN, DRew, show strong improvements in the deep regime, outperforming the others. This aligns with the intuition that eccentricity, being a node-level but globally-informed property, benefits from many message-passing layers to capture distant context, highlighting the strength of long-range architectures.

- In the case of `sssp` (Figure 8c) we again observe strong depth-related improvements, with the exception of GraphCON. Notably, SWAN, GPS, and DRew achieve large gains in the deep regime. Traditional models such as GCN and GIN or GraphCON exhibit plateau or degradation, revealing limited depth scalability.

- On the `ECHO-Charge` task (Figure 8d), the behavior differs. This task involves precise regression of atomic partial charges, where small errors matter. Most models show stable MAE across depths, except for GCN and GIN, which degrade significantly in the deep regime. Importantly, models with explicit long-range message-passing capabilities (A-DGN, SWAN, PH-DGN, and GPS) retain high accuracy even at $> 17$ layers. This suggests their robustness in fine-grained, long-range molecular prediction tasks.

- Finally, on the `ECHO-Energy` task (Figure 8e), performance trends are more model-dependent and exhibit higher variability, while still underlying the benefit of deeper regimes. Generally, non-dissipative architectures such as PH-DGN and A-DGN benefit from increased depth. The best performance is achieved by the transformer-based model GPS, supporting the intuition that this graph-level task requires globally informed representations.

Overall, the observed patterns reveal a clear correlation between the number of message-passing layers and performance: models require many layers to perform well, confirming that these benchmarks require information to propagate over long distances. Remarkably, architectures explicitly designed to support many message-passing steps consistently outperform others, further confirming the long-range nature of our proposed benchmarks.

### G.2    PERFORMANCE ACROSS GRAPH DIAMETERS

Figure 9 reports, for the best configuration of each model (as selected during model selection), the test MAE across varying graph diameters for all tasks. This analysis highlights how different models handle increasingly long-range dependencies.

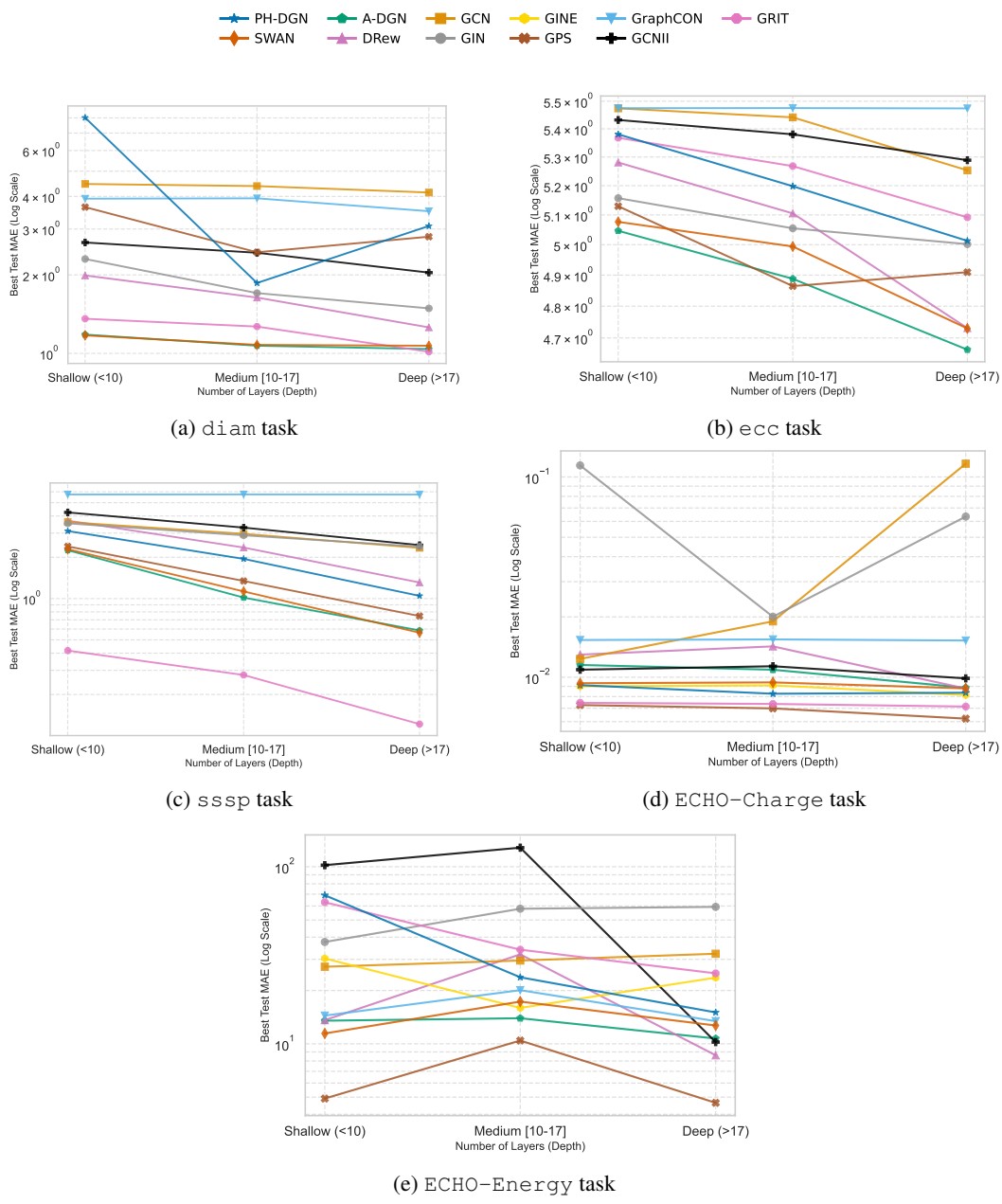

Figure 8: Test MAE at different numbers of GNN layers across tasks.

For the `diam` task (Figure 9a), most models show robust performance for small to moderate diameters, with a slight increase in MAE for very large diameters. Notably, GCN, GraphCON and GCNII architectures exhibit substantial degradation as diameter increases, suggesting poor scalability in capturing global structure on many message-passing steps. Again, non-dissipative architectures (i.e., A-DGN, PH-DGN, and SWAN), DRew, GPS, and GRIT remain consistently accurate across all graph diameters, demonstrating their capacity to generalize across different graph scales. Finally, we observe that models with lower generalization capabilities tend to bias their predictions towards the statistical mean of the dataset. Consequently, these models achieve low error rates in the central range, where the ground truth aligns with the mean, but show high errors at the tails, as they fail to distinguish graphs with extreme diameters from the average case.

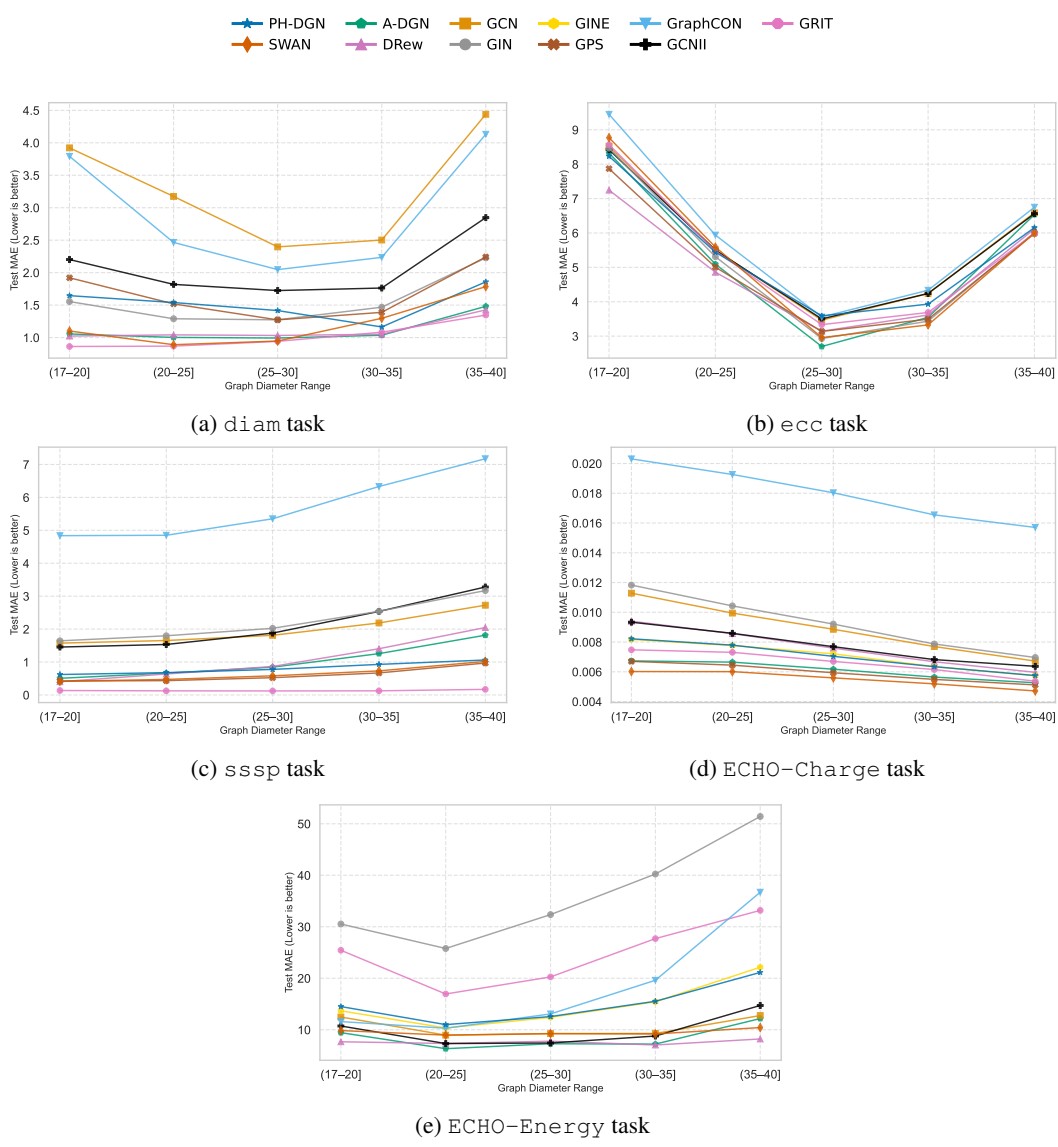

Figure 9: Test MAE for the best configuration of each model (as selected during model selection) at different graph diameters across synthetic and molecular tasks.

The `ecc` task (Figure 9b) reveals a characteristic U-shaped curve. Performance improves as diameter increases from small to moderate values, and deteriorates again for very large graphs. Here, all models follow a similar performance pattern that is highly correlated with the node eccentricity distribution depicted in Figure 4, creating an inverse relationship between sample frequency and error. In particular, in the center of the plot, it is possible to see the GNNs that learn the dominant pattern; in addition, simply predicting values close to the statistical mean minimizes the expected loss in this dense region, and results in a lower error. Conversely, graphs exhibiting lower eccentricities are statistically less frequent, leading to a noticeable degradation of the performance across extreme values.

In the `sssp` task (Figure 9c), increasing graph diameter consistently correlates with rising MAE. Model performance divides into three groups, with GraphCON exhibiting the worst performance both in terms of overall MAE and degradation with increasing diameter. GCN, GCNII, and GIN show similar values across the task and similar degradation trends. Finally, non-dissipative models, Graph Transformers, and DRew once again demonstrate remarkable and consistent performance even

on difficult graphs. This trend reinforces the long-range nature of the task, where deeper or more expressive models are required to maintain strong performance.

On the molecular `ECHO-Charge` task (Figure 9d), test MAE consistent across all ranges, but subtle trends emerge. Models like DRew and GPS show stability and even slight improvements for larger molecular graphs. Interestingly, GINE performs substantially better than its counterpart GIN, suggesting that edge-level attributes play a crucial role in the `ECHO-Charge` regression task.

On the molecular task `ECHO-Energy` (Figure 9e), a clear trend emerges: test MAE increases with graph diameter across all models. However, architectures such as SWAN and DRew exhibit substantially lower sensitivity to diameter, maintaining a nearly constant MAE across the full range. Additionally, we note that all models exhibit a general performance drop when processing molecular graphs with a diameter greater than 35. We attribute this behavior to the original ChEMBL dataset's distribution, which includes fewer graphs with diameters in the 35–40 range, as illustrated in Figure 6.

Overall, this complementary diameter-wise analysis underlines the necessity for architectures capable of handling variable and large receptive fields. It also highlights that while shallow models may perform competitively on small graphs, their limitations become apparent in regimes requiring long-range reasoning.

## G.3 IMPACT OF READOUT LAYER DEPTH

To examine the impact of the readout depth, we conducted an experiment varying the depth of the readout (from 1 to 3 layers) on the `ECHO-Synth` tasks. Table 17 presents the performance of a GCN with varying readout layers. The results indicate that increasing the readout depth to three layers does not improve GCN performance on the synthetic tasks. Therefore, the final performance appears to be independent of the readout depth. Since the additional layer does not yield measurable benefits, we consider the increased model complexity unjustified. For this reason, we retain a two-layer readout to prioritize computational efficiency and performance.

Table 17: Ablation study on Readout Depth for GCN on ECHO-Synth tasks. Metrics are reported as Mean Absolute Error (MAE) $\pm$ Standard Deviation.

| Model | diam $\downarrow$ | ecc $\downarrow$ | sssp $\downarrow$ |
|---|---|---|---|
| GCN (1 layer readout) | $6.219_{\pm 0.387}$ | $5.493_{\pm 0.007}$ | $2.367_{\pm 0.083}$ |
| GCN (2 layers readout) | $3.832_{\pm 0.262}$ | $5.233_{\pm 0.034}$ | $2.102_{\pm 0.094}$ |
| GCN (3 layers readout) | $5.743_{\pm 0.009}$ | $5.172_{\pm 0.091}$ | $2.075_{\pm 0.545}$ |

## G.4 PERFORMANCE ACROSS DIFFERENT GRAPH TOPOLOGIES

In this section, we analyze the models' performance on different graph topologies in `ECHO-Synth`. Figures 10, 11 and 12 show respectively `diam`, `ecc` and `sssp`. The results show that although absolute performance varies slightly with the underlying graph topology (e.g., GPS performs better on lobster graphs than on tree-like topologies in SSSP), the relative ranking of models remains consistent, reinforcing the robustness of our findings. Interestingly, we observe that the line topology is consistently the most challenging one. This is expected: in a line graph, every message must pass through a sequence of intermediate nodes, making each node a critical bottleneck for information propagation and amplifying the need for long-range communication.

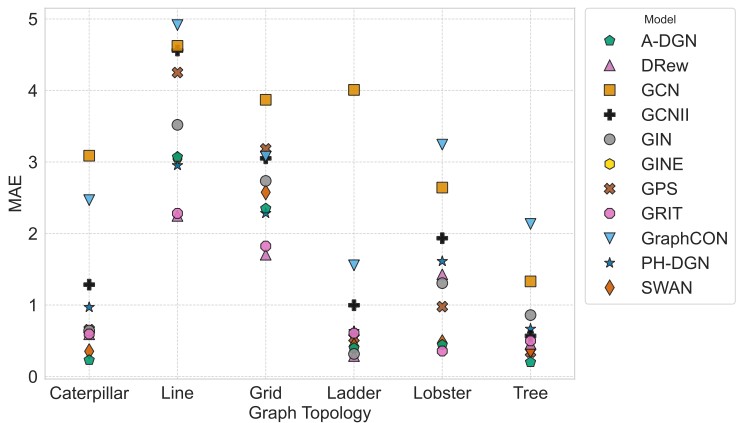

Figure 10: Results by different topologies on `diam` task.

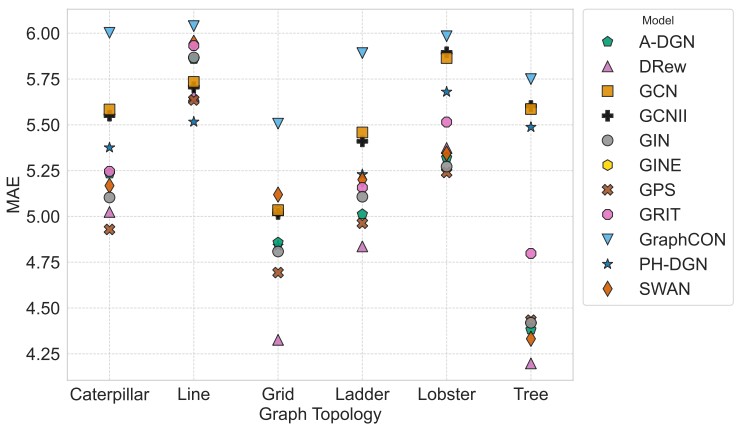

Figure 11: Results by different topologies on `ecc` task.

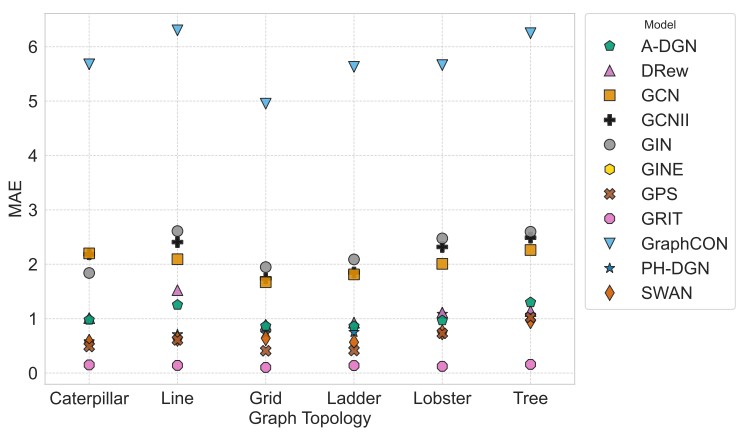

Figure 12: Results by different topologies on `sssp` task.

# H    EXTENDED RESULTS

In Table 18, we report additional results on `ECHO-Synth` benchmark. In particular, we report MAE, MSE and loss (defined as $\log_{10}(\text{MSE})$) obtained on the test set. Similarly, we report the same metrics for `ECHO-Charge` and `ECHO-Energy` in Table 19 and Table 20, respectively.

Table 18: Test performance (mean ± std) of different models across `ECHO-Synth` tasks. Lower is better. In bold the best model.

| Model | Test Loss ↓ | Test MSE ↓ | Test MAE ↓ |
|---|---|---|---|
| | | diam | |
| A-DGN | $-2.531_{\pm 0.010}$ | $4.818_{\pm 0.108}$ | $1.151_{\pm 0.038}$ |
| DRew | $\mathbf{-2.635_{\pm 0.020}}$ | $\mathbf{3.756_{\pm 0.170}}$ | $1.243_{\pm 0.047}$ |
| GCN | $-1.848_{\pm 0.051}$ | $22.872_{\pm 2.766}$ | $3.832_{\pm 0.262}$ |
| GCNII | $-2.227_{\pm 0.026}$ | $9.696_{\pm 0.568}$ | $2.005_{\pm 0.093}$ |
| GIN | $-2.356_{\pm 0.066}$ | $7.238_{\pm 1.153}$ | $1.630_{\pm 0.161}$ |
| GPS | $-2.192_{\pm 0.025}$ | $10.454_{\pm 0.610}$ | $2.160_{\pm 0.098}$ |
| GraphCON | $-1.995_{\pm 0.037}$ | $16.427_{\pm 1.419}$ | $2.969_{\pm 0.189}$ |
| GRIT | $-2.630_{\pm 0.038}$ | $3.877_{\pm 0.295}$ | $\mathbf{1.014_{\pm 0.046}}$ |
| PH-DGN | $-2.416_{\pm 0.181}$ | $6.699_{\pm 2.728}$ | $1.627_{\pm 0.398}$ |
| SWAN | $-2.517_{\pm 0.023}$ | $4.950_{\pm 0.265}$ | $1.121_{\pm 0.070}$ |
| | | ecc | |
| A-DGN | $-1.649_{\pm 0.006}$ | $35.967_{\pm 0.492}$ | $4.981_{\pm 0.037}$ |
| DRew | $\mathbf{-1.696_{\pm 0.002}}$ | $\mathbf{32.247_{\pm 0.148}}$ | $\mathbf{4.651_{\pm 0.020}}$ |
| GCN | $-1.606_{\pm 0.005}$ | $39.706_{\pm 0.460}$ | $5.233_{\pm 0.034}$ |
| GCNII | $-1.603_{\pm 0.006}$ | $39.911_{\pm 0.518}$ | $5.241_{\pm 0.030}$ |
| GIN | $-1.668_{\pm 0.015}$ | $34.454_{\pm 1.201}$ | $4.869_{\pm 0.092}$ |
| GPS | $-1.682_{\pm 0.003}$ | $33.346_{\pm 0.226}$ | $4.758_{\pm 0.021}$ |
| GraphCON | $-1.566_{\pm 0.001}$ | $43.505_{\pm 0.017}$ | $5.474_{\pm 0.001}$ |
| GRIT | $-1.618_{\pm 0.022}$ | $38.667_{\pm 1.903}$ | $5.091_{\pm 0.158}$ |
| PH-DGN | $-1.630_{\pm 0.017}$ | $37.510_{\pm 1.416}$ | $5.068_{\pm 0.126}$ |
| SWAN | $-1.671_{\pm 0.007}$ | $34.208_{\pm 0.578}$ | $4.840_{\pm 0.045}$ |
| | | sssp | |
| A-DGN | $-2.566_{\pm 0.089}$ | $4.425_{\pm 0.879}$ | $1.176_{\pm 0.140}$ |
| DRew | $-2.386_{\pm 0.001}$ | $6.589_{\pm 0.015}$ | $1.279_{\pm 0.011}$ |
| GCN | $-2.217_{\pm 0.033}$ | $9.743_{\pm 0.757}$ | $2.102_{\pm 0.094}$ |
| GCNII | $-2.213_{\pm 0.177}$ | $10.369_{\pm 3.575}$ | $2.128_{\pm 0.429}$ |
| GIN | $-2.138_{\pm 0.090}$ | $11.868_{\pm 2.689}$ | $2.234_{\pm 0.271}$ |
| GPS | $-3.115_{\pm 0.040}$ | $1.255_{\pm 0.113}$ | $0.472_{\pm 0.050}$ |
| GraphCON | $-1.488_{\pm 0.000}$ | $52.104_{\pm 0.016}$ | $5.734_{\pm 0.011}$ |
| GRIT | $\mathbf{-4.119_{\pm 0.264}}$ | $\mathbf{0.147_{\pm 0.083}}$ | $\mathbf{0.121_{\pm 0.013}}$ |
| PH-DGN | $-2.616_{\pm 0.317}$ | $4.656_{\pm 3.013}$ | $1.323_{\pm 0.485}$ |
| SWAN | $-2.782_{\pm 0.205}$ | $2.905_{\pm 1.556}$ | $0.896_{\pm 0.232}$ |

Table 19: Test performance (mean ± std) of different models across the `ECHO-Charge` task. Lower is better. In bold the best model. Test loss $(\log_{10}(\text{MSE}))$ is computed on the normalized dataset, while test MSE and test MAE are reported on the original (non-scaled) data.

| Model | Test Loss ↓ | Test MSE $\times 10^4$ ↓ | Test MAE $\times 10^3$ ↓ |
|---|---|---|---|
| A-DGN | $-3.840_{\pm 0.009}$ | $1.456_{\pm 0.032}$ | $6.543_{\pm 0.146}$ |
| DRew | $-3.444_{\pm 0.054}$ | $3.669_{\pm 0.459}$ | $9.086_{\pm 0.473}$ |
| GCN | $-3.508_{\pm 0.086}$ | $3.126_{\pm 0.263}$ | $8.421_{\pm 0.512}$ |
| GCNII | $-3.462_{\pm 0.019}$ | $3.490_{\pm 0.147}$ | $8.829_{\pm 0.021}$ |
| GIN | $-3.245_{\pm 0.038}$ | $5.750_{\pm 0.239}$ | $10.784_{\pm 0.059}$ |
| GINE | $-3.648_{\pm 0.020}$ | $2.284_{\pm 0.402}$ | $7.176_{\pm 0.371}$ |
| GPS | $-3.821_{\pm 0.018}$ | $1.620_{\pm 0.065}$ | $6.182_{\pm 0.219}$ |
| GraphCON | $-2.879_{\pm 0.009}$ | $13.25_{\pm 0.265}$ | $19.629_{\pm 0.195}$ |
| GRIT | $-3.762_{\pm 0.0177}$ | $1.765_{\pm 0.071}$ | $7.134_{\pm 6.090}$ |
| PH-DGN | $-3.595_{\pm 0.024}$ | $2.562_{\pm 0.144}$ | $7.915_{\pm 0.269}$ |
| SWAN | $\mathbf{-3.907_{\pm 0.027}}$ | $\mathbf{1.251_{\pm 0.029}}$ | $\mathbf{6.109_{\pm 0.103}}$ |

Table 20: Test performance (mean ± std) of different models across the `ECHO-Energy` task. Lower is better. In bold the best model. Test loss $(\log_{10}(\text{MSE}))$ is computed on the normalized dataset, while test MSE and test MAE are reported on the original (non-scaled) data.

| Model | Test Loss ↓ | Test MSE $\times 10^3$ ↓ | Test MAE $\times 10^2$ ↓ |
|---|---|---|---|
| A-DGN | $-4.857_{\pm 0.083}$ | $1.415_{\pm 0.799}$ | $0.125_{\pm 0.016}$ |
| DRew | $-5.007_{\pm 0.231}$ | $1.281_{\pm 0.733}$ | $0.113_{\pm 0.024}$ |
| GCN | $-4.210_{\pm 0.010}$ | $4.561_{\pm 0.176}$ | $0.281_{\pm 0.012}$ |
| GCNII | $-4.884_{\pm 0.196}$ | $1.560_{\pm 0.653}$ | $0.132_{\pm 0.026}$ |
| GIN | $-3.800_{\pm 0.160}$ | $12.215_{\pm 2.878}$ | $0.479_{\pm 0.102}$ |
| GINE | $-4.418_{\pm 0.265}$ | $5.225_{\pm 2.536}$ | $0.236_{\pm 0.076}$ |
| GPS | $\mathbf{-5.786_{\pm 0.118}}$ | $\mathbf{0.180_{\pm 0.045}}$ | $\mathbf{0.053_{\pm 0.008}}$ |
| GRIT | $-4.348_{\pm 0.084}$ | $5.122_{\pm 1.492}$ | $0.255_{\pm 0.025}$ |
| GraphCON | $-4.817_{\pm 0.089}$ | $0.975_{\pm 0.242}$ | $0.143_{\pm 0.008}$ |
| PH-DGN | $-4.717_{\pm 0.046}$ | $1.359_{\pm 0.408}$ | $0.161_{\pm 0.011}$ |
| SWAN | $-4.825_{\pm 0.107}$ | $2.652_{\pm 2.257}$ | $0.126_{\pm 0.012}$ |

# I    RUNTIMES

To assess the computational efficiency and predictive performance of all models, we report both training and inference runtimes measured on a NVIDIA H100 GPU, as well as the mean absolute error (MAE) across tasks in the `ECHO` benchmark (see Table 21). Training time is measured as the average per-epoch duration over 10 epochs, while inference time is computed as the average forward pass duration over 10 independent runs on the test set, using a batch size of 512. The three metrics correspond to the best hyperparameter configuration selected for each model. This comprehensive evaluation allows a direct comparison of models not only in terms of accuracy but also with respect to their scalability and practical deployability. We note that DRew's reported runtime does not include the preprocessing step, which involves computing the Floyd–Warshall algorithm (Cormen et al., 2009), a procedure with cubic time complexity in the number of nodes.

Table 21 highlights that while transformer-based models like GPS achieve strong performance on long-range tasks, particularly on the real-world `ECHO-Charge`/`ECHO-Energy` dataset, they do so at the cost of significantly higher computational overhead. In contrast, architectures such as SWAN and A-DGN strike a more favorable balance between efficiency and accuracy, suggesting the potential of non-dissipative DE-GNNs in overcoming the limitations of standard message passing.

Table 21: Training and inference runtime (in seconds, mean $\pm$ standard deviation) on the ECHO Benchmark. Results for ECHO-Synth were measured on an NVIDIA H100 GPU, while ECHO-Chem tasks were measured on an NVIDIA L40S GPU. Training time refers to the average time per epoch computed over 10 epochs. Inference time refers to the forward pass on the test set, computed over 10 independent runs. In both cases the batch size is set to 256. For each task, the reported values correspond to the best configuration of each model as selected during model selection. To ease comparison, we also report the performance of each model alongside its runtime. Note that ECHO-Charge MAE values should be multiplied by $\times 10^{-3}$ for correct interpretation. DRew's reported runtime does not include the pre-processing step, which involves computing the Floyd–Warshall algorithm, a procedure with cubic time complexity in the number of nodes.

| Metric | Model | ECHO-Synth | | | ECHO-Chem | |
|---|---|---|---|---|---|---|
| | | diam | sssp | ecc | ECHO-Charge | ECHO-Energy |
| Training (s) | A-DGN | $1.430_{\pm 0.100}$ | $1.460_{\pm 0.130}$ | $1.710_{\pm 0.070}$ | $12.847_{\pm 0.543}$ | $36.476_{\pm 0.409}$ |
| Inference (s) | | $0.028_{\pm 0.002}$ | $0.018_{\pm 0.001}$ | $0.027_{\pm 0.001}$ | $0.191_{\pm 0.193}$ | $2.124_{\pm 0.197}$ |
| MAE | | $1.151_{\pm 0.038}$ | $1.176_{\pm 0.140}$ | $4.981_{\pm 0.037}$ | $6.543_{\pm 0.146}$ | $12.486_{\pm 1.621}$ |
| Training (s) | DRew | $1.920_{\pm 0.050}$ | $1.880_{\pm 0.060}$ | $1.760_{\pm 0.100}$ | $17.648_{\pm 0.325}$ | $17.318_{\pm 1.061}$ |
| Inference (s) | | $0.100_{\pm 0.001}$ | $0.043_{\pm 0.001}$ | $0.057_{\pm 0.002}$ | $0.606_{\pm 0.557}$ | $0.847_{\pm 0.563}$ |
| Pre-processing (s) | | $48.108_{\pm 0.943}$ | $48.108_{\pm 0.943}$ | $48.108_{\pm 0.943}$ | $345.379_{\pm 1.592}$ | $404.337_{\pm 1.601}$ |
| MAE | | $1.243_{\pm 0.047}$ | $1.279_{\pm 0.011}$ | $\mathbf{4.651_{\pm 0.020}}$ | $9.086_{\pm 0.473}$ | $11.325_{\pm 2.394}$ |
| Training (s) | GCNII | $1.700_{\pm 0.050}$ | $1.830_{\pm 0.020}$ | $1.620_{\pm 0.110}$ | $19.013_{\pm 0.286}$ | $23.039_{\pm 0.89}$ |
| Inference (s) | | $0.071_{\pm 0.002}$ | $0.062_{\pm 0.001}$ | $0.059_{\pm 0.001}$ | $0.829_{\pm 0.728}$ | $1.466_{\pm 0.982}$ |
| MAE | | $2.005_{\pm 0.093}$ | $2.128_{\pm 0.429}$ | $5.241_{\pm 0.030}$ | $8.829_{\pm 0.021}$ | $13.235_{\pm 2.630}$ |
| Training (s) | GCN | $1.480_{\pm 0.060}$ | $1.790_{\pm 0.060}$ | $1.450_{\pm 0.100}$ | $9.418_{\pm 0.388}$ | $8.976_{\pm 0.49}$ |
| Inference (s) | | $0.048_{\pm 0.001}$ | $0.065_{\pm 0.003}$ | $0.046_{\pm 0.002}$ | $0.376_{\pm 0.472}$ | $0.385_{\pm 0.388}$ |
| MAE | | $3.832_{\pm 0.262}$ | $2.102_{\pm 0.004}$ | $5.233_{\pm 0.034}$ | $8.421_{\pm 0.512}$ | $28.112_{\pm 1.239}$ |
| Training (s) | GIN | $1.410_{\pm 0.220}$ | $1.370_{\pm 0.060}$ | $1.340_{\pm 0.040}$ | $9.066_{\pm 0.509}$ | $10.259_{\pm 0.471}$ |
| Inference (s) | | $0.020_{\pm 0.001}$ | $0.019_{\pm 0.001}$ | $0.016_{\pm 0.001}$ | $0.065_{\pm 0.043}$ | $1.305_{\pm 0.109}$ |
| MAE | | $1.630_{\pm 0.161}$ | $2.234_{\pm 0.271}$ | $4.869_{\pm 0.092}$ | $10.784_{\pm 0.059}$ | $47.851_{\pm 10.154}$ |
| Training (s) | GINE | N/A | N/A | N/A | $18.978_{\pm 0.778}$ | $13.615_{\pm 0.619}$ |
| Inference (s) | | N/A | N/A | N/A | $0.138_{\pm 0.003}$ | $1.311_{\pm 1.834}$ |
| MAE | | N/A | N/A | N/A | $7.176_{\pm 0.371}$ | $23.558_{\pm 7.568}$ |
| Training (s) | GPS | $9.720_{\pm 0.070}$ | $14.580_{\pm 0.210}$ | $11.960_{\pm 0.050}$ | $788.985_{\pm 0.166}$ | $383.794_{\pm 1.975}$ |
| Inference (s) | | $4.536_{\pm 0.006}$ | $7.026_{\pm 0.001}$ | $6.235_{\pm 0.076}$ | $34.264_{\pm 2.416}$ | $89.699_{\pm 5.476}$ |
| MAE | | $2.160_{\pm 0.098}$ | $0.472_{\pm 0.050}$ | $4.758_{\pm 0.021}$ | $6.182_{\pm 0.219}$ | $\mathbf{5.257_{\pm 0.842}}$ |
| Training (s) | GraphCON | $0.990_{\pm 0.120}$ | $0.920_{\pm 0.040}$ | $0.940_{\pm 0.190}$ | $7.471_{\pm 0.429}$ | $9.037_{\pm 0.79}$ |
| Inference (s) | | $0.006_{\pm 0.001}$ | $0.004_{\pm 0.001}$ | $0.006_{\pm 0.001}$ | $0.146_{\pm 0.218}$ | $1.089_{\pm 0.334}$ |
| MAE | | $2.969_{\pm 0.189}$ | $5.734_{\pm 0.011}$ | $5.474_{\pm 0.001}$ | $19.629_{\pm 0.195}$ | $14.295_{\pm 0.807}$ |
| Training (s) | GRIT | $2.136_{\pm 0.098}$ | $2.898_{\pm 0.663}$ | $2.150_{\pm 0.533}$ | $55.964_{\pm 0.962}$ | $35.246_{\pm 0.539}$ |
| Inference (s) | | $0.579_{\pm 0.064}$ | $0.703_{\pm 0.175}$ | $0.632_{\pm 0.132}$ | $1.474_{\pm 0.958}$ | $1.940_{\pm 0.435}$ |
| MAE | | $\mathbf{1.014_{\pm 0.046}}$ | $\mathbf{0.121_{\pm 0.013}}$ | $5.091_{\pm 0.158}$ | $7.134_{\pm 6.090}$ | $25.508_{\pm 2.507}$ |
| Training (s) | PH-DGN | $2.840_{\pm 0.060}$ | $4.480_{\pm 0.060}$ | $3.010_{\pm 0.060}$ | $39.405_{\pm 1.753}$ | $43.653_{\pm 1.199}$ |
| Inference (s) | | $0.180_{\pm 0.011}$ | $0.375_{\pm 0.002}$ | $0.299_{\pm 0.006}$ | $1.218_{\pm 0.535}$ | $2.311_{\pm 0.560}$ |
| MAE | | $1.627_{\pm 0.398}$ | $1.323_{\pm 0.485}$ | $5.068_{\pm 0.126}$ | $7.915_{\pm 0.269}$ | $16.080_{\pm 1.123}$ |
| Training (s) | SWAN | $2.330_{\pm 0.120}$ | $2.130_{\pm 0.050}$ | $2.090_{\pm 0.110}$ | $54.528_{\pm 0.357}$ | $29.774_{\pm 0.421}$ |
| Inference (s) | | $0.203_{\pm 0.002}$ | $0.099_{\pm 0.001}$ | $0.168_{\pm 0.001}$ | $0.595_{\pm 0.520}$ | $1.873_{\pm 0.101}$ |
| MAE | | $1.121_{\pm 0.070}$ | $0.896_{\pm 0.232}$ | $4.840_{\pm 0.045}$ | $\mathbf{6.109_{\pm 0.103}}$ | $12.629_{\pm 0.807}$ |

## J VISUALIZATION OF GPS ATTENTION PATTERNS

In this section, we analyze the attention patterns of the GPS model within the sssp task. These patterns are illustrated in Figure 13. We observed that, starting from the first layer, the highest attention scores are often assigned to pairs of nodes that are not directly connected and that are usually far apart in the underlying graph. Notably, the model appears to identify one or a few nodes as central hubs that aggregate and redistribute information from these distant nodes. This mechanism effectively reduces the maximal traversal distance to only few hops, allowing distant nodes to communicate more easily. Therefore, this mechanism effectively reveals how the model routes long-range communication through structural shortcuts, thus confirming the long-range nature of the proposed tasks.

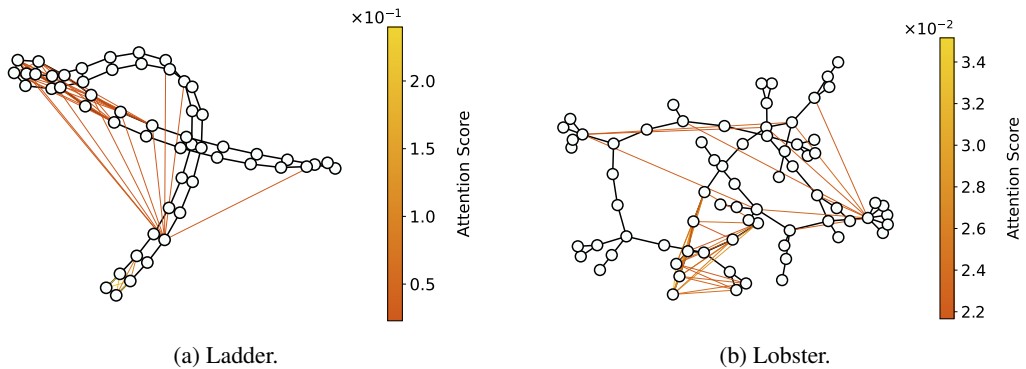

(a) Ladder.

(b) Lobster.

Figure 13: Visualization of the first-layer GPS attention scores in `sssp` for the Ladder and Lobster topologies, averaged across all heads. The top 40 attention-weighted node pairs are highlighted in color, while the original graph topology is shown in black.

