# OpenReview forum: "Can You Hear Me Now? A Benchmark for Long-Range Graph Propagation"
_ICLR.cc/2026/Conference — ICLR 2026 Poster_

### Official Review · Reviewer_DTw9 · 2025-10-26

**Soundness:** 4
**Presentation:** 3
**Contribution:** 3
**Rating:** 8
**Confidence:** 4

**Summary:**

This paper introduces a new benchmark for long-range graph propagation in graph learning, motivated by the fact that existing benchmarks (as the LRGB) have recently been questioned to actually capture long-range ability and somewhat plateaued. The authors introduce ECHO, consisting of a synthetic and real-world part. The synethetic tasks consists of single-source shortest path, node eccentricity, and graph diameter, and are to be predicted on graphs sampled from 6 different (sparse and high-diameter) graph topologies. The real-world tasks on molecules are predicting atomic charges (ECHO-Charge) and total energy (ECHO-Energy). The new benchmark is evaluated on a wide range of common graph learning methods, including standard MPNNs and graph transformers. Results indicate that models with global information exchange are performing better on ECHO.

**Strengths:**

- I personally think this is an important and timely contribution. I think that the graph ML community urgently needs new and well-designed long-range propagation benchmarks, as performance on the LRGB has plataued and been questioned to really probe long range propagation. Yet, most current methodological works still use the LRGB, especially the peptides tasks, often as the sole empirical basis for any claims regarding long-range ability.
- The benchmark design is quite good (synthetic and real-world), and it is clear that significant thought and computational resources went into the creation and evaluation on the baselines. For example, the benchmark is large-scale (200k graphs for the ECHO-Charge/Energy), and in the synthetic case, the graph topologies are properly stratified in train/val/test.
- The experimental rigor is high. Hyperparameters were extensively tuned on the benchmark (app. F), which is crucial given how untuned baselines have led to incorrect conclusions in the past.

**Weaknesses:**

- The graphs in all tasks are quite small (order of roughly 50-200 nodes). The paper could be more upfront about the benchmark not probing scalability, even though many current methodological works target methods that both capture long-range interactions and scale well. It might also help to explicitly position the work against newer long-range benchmarks that explicitly test scalability. [1]
- Not really a weakness but a suggestion: For the synthetic tasks, readers might initially think that the targets are simple monotonic functions of graph size within each topology, reducing the problem to a topology classification. I checked the data and this does not seem to be the case, but additionally grouping plots like Fig. 6(a) by topology could better convey task difficulty and make the reported MAE values more interpretable.

[1] Huidong Liang, Haitz Sáez de Ocáriz Borde, Baskaran Sripathmanathan, Michael Bronstein, Xiaowen Dong. *Towards Quantifying Long-Range Interactions in Graph Machine Learning: a Large Graph Dataset and a Measurement.* https://arxiv.org/pdf/2503.09008

**Questions:**

- Could you please refer to the points from “Weaknesses”?
- Do you have any intuition for the U-shaped error in fig. 8(b), and, to some extent, (a)?
- Superficially, ECHO-Synth resembles the graph property prediction (GPP) dataset from [2] (same tasks but different graph topologies).  Since [2] is cited, it would help to clarify the relationship, i.e., if/how ECHO-Synth builds on or departs from GPP, more explicitly (perhaps in sec. 3.1).

[2] Alessio Gravina, Davide Bacciu, and Claudio Gallicchio. Anti-Symmetric DGN: a stable architecture for Deep Graph Networks. ICLR 2023.

---

> ### Author Response · Authors · 2025-11-21
> **Response pt. 1**
>
> We sincerely thank the Reviewer for their thoughtful and encouraging evaluation of our work. We are very grateful for the recognition of ECHO as a timely and important contribution addressing the community’s growing need for reliable long-range propagation benchmarks, especially in light of the limitations of existing datasets such as LRGB.
> We appreciate the Reviewer’s positive comments on the benchmark design, which combines synthetic with real-world molecular tasks. We also thank the Reviewer for noting the scale and care of our dataset construction and the stratified train/validation/test splits used in the synthetic benchmarks to ensure fair evaluation.
> Finally, we are grateful that the Reviewer highlighted our rigorous experimental setup, including the extensive hyperparameter tuning. Below we address the Reviewer's comments.
>
> W1)
>
> We thank the Reviewer for the insightful comment. We would like to point out that our proposed datasets already position beyond the scale of most existing benchmarks in the literature. Increasing graph sizes to tens of thousands of nodes would raise significant scalability challenges and limit accessibility, as only groups with large computational resources (multiple powerful GPUs) would be able to run the benchmark. Moreover, this would raise sustainability concerns due to the high computational cost involved in testing. We emphasize that our goal was to provide a benchmark that is practical, accessible, and already extends beyond current standards, striking a balance between long-range complexity and usability by the broader community, avoiding digital divide concerns while still reflecting real-world scientific challenges. Nonetheless, we view ECHO and City-Networks [1] (which we consider contemporaneous with our work as it was accepted to NeurIPS 2025 after the submission of our paper) as complementary efforts addressing long-range propagation problems. Specifically, [1] focuses on one single synthetic task on large graphs. Differently, our ECHO consists of 5 (3 synthetic and 2 real-world) tasks, which are inherently long-range and computationally accessible, providing a controlled setting to assess propagation capabilities. Since we believe this discussion can better position our paper, we have included this discussion in the revised paper (Section 2).
>
>
> W2)
>
> Following the Reviewer suggestion (also proposed by Reviewer 5FWW), we have added an ablation in Appendix G.4 to make per-topology performance differences explicit and improve interpretability of MAE values in our results. The results show that although absolute performance varies slightly with the underlying graph structure (e.g., GPS performs better on lobster graphs than on tree-like topologies in SSSP), the relative ranking of models remains consistent, reinforcing the robustness of our findings. Interestingly, we observe that the line topology is consistently the most challenging topology. This is expected: in a line graph, every message must pass through a sequence of intermediate nodes, making each node a critical bottleneck for information propagation and amplifying the need for long-range communication. We thank again the Reviewer for the suggestion.

---

> > ### Author Response · Authors · 2025-11-21
> > **Response pt. 2**
> >
> > Q1)
> >
> > See W1, W2
> >
> > Q2)
> >
> > We thank the Reviewer for highlighting the distinct 'U-shape' observed in the Test MAE plots (Fig. 9b). We agree that this trend merits further clarification. We have determined that this U-shaped error curve is a direct consequence of the non-uniform distribution of eccentricity values within the synthetic dataset, which creates an inverse relationship between sample frequency and error. In particular:
> >  - Center (Low MAE): Here, the GNNs effectively learn this dominant pattern. Furthermore, simply predicting values close to the statistical mean minimizes the expected loss in this dense region, naturally resulting in a lower error.
> > - Tails (High MAE): Conversely, at the tails, graphs exhibiting extreme eccentricity values are statistically less frequent.  Due to this, the models have lower generalization ability. This bias towards the mean explains the U-shape observed in the chart.
> >
> > A similar interpretation applies to Fig. 9a. Models with lower generalization capabilities (i.e., GCN and GraphCON in the diameter task) tend to bias their predictions towards the statistical mean of the dataset. Consequently, these models achieve low error rates in the central range, where the ground truth aligns with the mean, but show high errors at the tails, as they fail to distinguish graphs with extreme diameters from the average case.
> > We have included this discussion in the Appendix G.2.
> >
> > Q3)
> >
> > We thank the Reviewer for the observation. ECHO-Synth indeed draws inspiration from the GPP benchmark [2], which extends the tasks 1, 2, and 5 in [3], and we now clarify this relationship more explicitly in the revised paper, Section 3.1.
> > While ECHO-Synth draws inspiration from [2,3], it departs from it in a key aspect. Both [2] and [3] use small graphs  (25–35 nodes in [2]) mostly sampled from distributions that yield highly connected structures with small diameters and limited long-range dependencies. In contrast, ECHO-Synth is explicitly designed to rigorously stress-test long-range capabilities. To this end, it (i) uses larger graphs (≈84 nodes) and (ii) employs topologies deliberately designed to introduce bottlenecks, increasing the graph diameter (up to 40 hops) and requiring substantially more propagation steps for accurate predictions. These design choices significantly increase the long-range difficulty of the tasks, making ECHO-Synth a more rigorous benchmark for evaluating long-range propagation than [2] and [3].
> >
> > ---
> > [1] Liang et al. Towards Quantifying Long-Range Interactions in Graph Machine Learning: a Large Graph Dataset and a Measurement.
> >
> > [2] Gravina et al. Anti-Symmetric DGN: a stable architecture for Deep Graph Networks. ICLR 2023.
> >
> > [3] Corso et al. "Principal neighbourhood aggregation for graph nets." Advances in neural information processing systems 33 (2020): 13260-13271.

---

> ### Comment · Reviewer_DTw9 · 2025-11-22
>
> I thank the authors for their thoughtful rebuttal, and all my questions have been answered. With the new updates, the work is now clearly positioned against City-Networks and ECHO-Synth against GPP. Just as a quick remark, I am not sure I agree with this statement:
>
> > Increasing graph sizes to tens of thousands of nodes would raise significant scalability challenges and limit accessibility, as only groups with large computational resources (multiple powerful GPUs) would be able to run the benchmark. Moreover, this would raise sustainability concerns due to the high computational cost involved in testing. We emphasize that our goal was to provide a benchmark that is practical, accessible, and already extends beyond current standards, striking a balance between long-range complexity and usability by the broader community, avoiding digital divide concerns while still reflecting real-world scientific challenges.
>
> Benchmarks with tens of thousands of nodes, e.g., PubMed  (19k) or ogbn-arxiv (170k) are routinely trained on a single 16–40 GB GPU which is standard academic compute, so I don't think that this poses any significant accessibility limitations. My original point was just that designing methods that can handle long-range interactions *but* also scale well (i.e. in situations where global attention is not feasible anymore) is also an highly active area and ECHO should be upfront about not testing this.
>
> However, since the revision now makes this clearer, I maintain my positive view and leave my score unchanged.

---

> > ### Author Response · Authors · 2025-11-25
> >
> > We thank the Reviewer for the prompt response and for maintaining the positive assessment.
> >
> > Regarding the remark on scalability: We agree that standard academic GPUs can easily handle single graphs of 20k+ nodes (like PubMed or ogbn-arxiv). However, the distinction lies in the dataset structure; while those are typically single-graph datasets, ECHO comprises up to 200k distinct graphs. Scaling each of these 200k graphs to tens of thousands of nodes would result in a total computational footprint that could exceed standard academic resources.
> >
> > We again thank the Reviewer for supporting the acceptance of our work.

---

### Official Review · Reviewer_5FWW · 2025-10-27

**Soundness:** 3
**Presentation:** 4
**Contribution:** 3
**Rating:** 6
**Confidence:** 3

**Summary:**

The paper introduces a suite of new inductive benchmark datasets specifically designed to evaluate long-range propagation capabilities in graph neural networks (GNs). The benchmark collection includes five tasks, of which three are synthetic and two are real world. The synthetic tasks consist of predicting property predictions on different topologies, two of which are node-level and one is graph-level. The real-world tasks consist of chemically grounded tasks, one being a node-level regression task where the goal is to predict atom charges, and the other is graph-level where the goal is to predict total energy. The paper argues that these datasets are indeed long-range. The benchmark is intended to fill the gap left by prior datasets (e.g. LRGB) as they received criticism recently for poor hyperparameter tuning and their long-range property having been questioned.

**Strengths:**

1. The paper is very well written and very clear.
2. The paper addresses an open problem consisting of finding a good long range benchmark for GNNs, which is particularly relevant given the recent criticisms of LRGB (hyperparameter tuning, and long-range nature of the task).
3. The hyperparameter tuning is more rigorous than previous work, directly addressing a criticism of previous benchmarks.
4. The ECHO-Charge and ECHO-Synth is original and may foster more work applying GNNs to chemistry applications.
5. The paper verifies the long-range nature of the ECHO-synth benchmarks by showing that the performance improves with number of layer.

**Weaknesses:**

1. **Long range evidence for real world tasks is lacking** The evidence for the long-range nature of the real world benchmark (ECHO-Charge and ECHO-Energy) is limited. Indeed there seems to be no correlation between depth and performance (Fig 7d). The only argument is that long-range architectures (such as GPS and SWAN) outperform standard supposedly not long range architectures. This argument is very similar to the LRGB arguments which were shown to be limited. On the other hand, the hyperparameter tuning in this work is more thorough.
2. **Hyperparameter tuning can be improved** The main criticism of LRGB in Tönshoff et al is that increasing the MLP depth post-readout improves performance on the proposed tasks. However, in this work the post-readout depth is set to 2 without tuning. In light of this, more thorough look at this hyperparameter would be interesting to the community and would further improve the hyperparameter tuning.
3. **[Minor] better evaluation of different topologies** In the synthetic datasets, the motivation for all different topologies are for example to include bottlenecks, however performance per graph topology was not evaluated.

**Questions:**

1. GPS is an architecture that combines several components: positional encoding, MPNN layers, and fully connected attention. Could you please add more information on what components were used and tuned (especially what MPNN architecture and number layers)?
2. In the synthetic datasets it says that the node features are random uniform. Wy not set the node features to zero?
3. Could you do an ablation on the post-pooling MLP depth for the graph-level real world task?

---

> ### Author Response · Authors · 2025-11-21
> **Response pt. 1**
>
> We sincerely thank the Reviewer for their thoughtful reading and positive evaluation of our work. We greatly appreciate the recognition of the clarity of our presentation, the relevance of developing a rigorous long-range benchmark for GNNs, and the originality of our tasks. We are also grateful that the Reviewer highlighted our careful hyperparameter tuning and empirical validation of long-range behavior. Below, we address the Reviewer’s comments point-by-point in our responses. We hope our clarifications and revisions will address your concerns and prompt a reconsideration of your evaluation and score.
>
> W1)
>
> The theoretical evidence that our molecular tasks are long-range is supported by chemico-physical evidence, such as in \[1,2,3\], which show that distant atoms can significantly alter charge distributions and total energy via long-range electrostatics and charge transfer. Specifically, the total molecular energy is computed considering several quantum-mechanical long-range interactions \[1\], and, similarly, the partial atomic charges are influenced by non-local electronic effects \[2,3\]. Therefore, this discussion, together with our empirical results (showing that model performance improves with larger numbers of message-passing steps) indicates that both ECHO-Charge and ECHO-Energy inherently require modeling interactions over large hops. Finally, we note that in Fig. 7d (which in the revised paper is 8d), GIN and GCN are the only models that do not exhibit performance gains at larger depths, consistent with their limited effectiveness on these tasks. All other models benefit from increased message-passing steps, further reinforcing our claim that capturing information from distant nodes is essential in these tasks. We included this discussion in the revised paper, Section 3.2, to clarify the intrinsic long-rangeness of our benchmark, and added the required references. Thank you.
>
> W2)
>
> Regarding the readout depth, we agree with the Reviewer that this is an interesting study and we now add an experiment on the ECHO-Synth tasks in Appendix G.3. In the table below, we report the results for your convenience.
>
> | Model         | Diam ↓| Ecc ↓       | SSSP ↓ |
> |---------------|---------------|------------|------------|
> |GCN (1 layers readout) | 6.219 ± 0.387   |   5.494 ± 0.007  |  2.367 ± 0.083 |
> |GCN (2 layers readout) | 3.832 ± 0.262   |   5.233 ± 0.034  |  2.102 ± 0.094 |
> |GCN (3 layers readout) | 5.743 ± 0.009    |   5.172 ± 0.091  |  2.075 ± 0.545 |
>
> The results indicate that increasing the readout depth to three layers does not improve GCN performance on the synthetic tasks. Therefore, the final performance appears to be independent of the readout depth. Since the additional layer does not yield measurable benefits, we consider the increased model complexity unjustified. For this reason, we retain a two-layer readout to prioritize computational efficiency and performance. We thank the Reviewer again for the insightful suggestion.
>
> W3 & Q3)
>
> We thank the Reviewer for the insightful suggestion. Following this, we have included an ablation study analyzing model performance across different graph topologies in ECHO-Synth, See Appendix G.4. The results show that although absolute performance varies slightly with the underlying graph structure (e.g., GPS performs better on lobster graphs than on tree-like topologies in SSSP), the relative ranking of models remains consistent, reinforcing the robustness of our findings. Interestingly, we observe that the line topology is consistently the most challenging topology. This is expected: in a line graph, every message must pass through a sequence of intermediate nodes, making each node a critical bottleneck for information propagation and amplifying the need for long-range communication.

---

> > ### Author Response · Authors · 2025-11-21
> > **Response pt. 2**
> >
> > (Q1)
> >
> > In our experiments, each GPS layer combines a one-layer GCN backbone with fully connected multi-head attention (with 2 attention heads). We have clarified these implementation details and hyperparameter settings in the revised manuscript (Table 12).
> >
> > (Q2)
> >
> > We thank the Reviewer for the question. We opted for random uniform node features rather than zero vectors for two reasons: (i) introduce stochasticity that makes the synthetic tasks less trivial, and (ii) to provide a unique identifier for the nodes in the tasks and prevent the trivial scenario in which all nodes share identical initial representations. This can hinder the expressiveness of certain architectures and make it difficult to observe meaningful differences in their behavior, as also discussed in \[4\]. We clarified this aspect in the manuscript in Section 3.1.
> >
> > (Q3)
> >
> > See Response pt. 1, [W3 & Q3]
> >
> > ---
> >
> > \[1\] Jensen. “Introduction to Computational Chemistry”. John Wiley & Sons. 2017
> >
> > \[2\] Ko, et al. "A fourth-generation high-dimensional neural network potential with accurate electrostatics including non-local charge transfer." Nature communications 12.1 (2021): 398\.
> >
> > \[3\] Shaidu, et al. "Incorporating long-range electrostatics in neural network potentials via variational charge equilibration from shortsighted ingredients." npj Computational Materials 10.1 (2024): 47\.
> >
> > \[4\] Sato, et al. Random Features Strengthen Graph Neural Networks. In Proceedings of the 2021 SIAM International Conference on Data Mining (SDM), 2021

---

> > > ### Comment · Reviewer_5FWW · 2025-11-24
> > >
> > > I thank the authors for their thorough rebuttal. My concerns are addressed, I have raised my score accordingly.
> > >
> > > A note on Q2, do you expect the results to change if the node features were constant? I think that there are arguments for keeping them constant as random node identifier are not standard practice in real world applications. I would be curious to the result of this experiment.

---

> > > > ### Author Response · Authors · 2025-11-25
> > > >
> > > > We sincerely thank the Reviewer for the thorough engagement with our rebuttal and for raising the score, supporting the acceptance of our work. Regarding Q2, we conducted an additional experiment using A-DGN (one of the top-performing models in our benchmark) on the ECHO-Synth dataset. The results are reported below:
> > > >
> > > > | Model         | Diam ↓| SSSP ↓ | ECC ↓ |
> > > > |---------------|---------------|------------| -----------|
> > > > |A-DGN (const. features set to 1) | 1.167 ± 0.058  |  2.829 ± 0.345  | 5.055 ± 0.035 |
> > > > |A-DGN (const. features set to 0) | 1.168 ± 0.187  |  1.915 ± 0.423  | 4.998 ± 0.036 |
> > > > |A-DGN (paper random node features)     | **1.151 ± 0.038**  |  **1.176 ± 0.140**  | **4.981 ± 0.037** |
> > > >
> > > > These results show that constant node features cause a performance drop in SSSP, while Diameter and ECC remain largely unaffected. Our intuition is that random node features are more useful in the SSSP task since they could help in better distinguishing the source nodes of the shortest paths.

---

### Official Review · Reviewer_5Ph3 · 2025-10-28

**Soundness:** 3
**Presentation:** 4
**Contribution:** 4
**Rating:** 8
**Confidence:** 4

**Summary:**

- The paper argues the need for newer and better-motivated long-range benchmarks than those currently in use by the community
- The paper introduces a new benchmark, ECHO, for long-range interactions in GNNs. The benchmark includes:
	- Echo-Synth: three synthetic tasks (two node level and one graph level), requiring the computation of shortest-path-based properties
		- Long-range dependency is assured as these properties require transversing the entire graph, and all synthetic graphs have diameters of at least 17 hops
	- Two real-world tasks based on atomic partial charges; Echo-Charge, a node-level regression of the partial charge of each atom in a molecular graph, and Echo-Energy, graph-level regression of total molecular energy
		- Long-range dependency is again ensured by graphs with minimum diameters of 17 hops, and argued based on the nature of the underlying task
- The paper includes a comprehensive suite of experiments on a fairly comprehensive range of architectures, including classical MPNNs, a multi-hop MPNN, differential-equation-inspired GNNs and a graph Transformer.

**Strengths:**

- The need for new long-range benchmarks and the shortcomings of existing ones are well-argued.
- The tasks, both synthetic and real-world, are at least as well motivated as existing benchmarks; I can see ECHO becoming a new and sorely-needed standard benchmark for the community.
- The paper is very well written, and Figures 1,2 and 3 are used to great effect to illustrate the tasks introduced; I think this will be a great boon to adoption.
- The authors evidently put a lot of time and effort into this paper; in addition to the experiments in the main text, the comprehensive additional experiments in the Appendix, and apparent 2 months of compute time required to produce the -Charge and -Energy datasets are evidence of this.
- I did not run the code, but it appears thorough, looks easy to run and is well-documented.
- I did not closely read the Appendix, but the inclusion of runtime measurements is welcome, as is the illustration of effective hub nodes in GPS, and the large depth and large diameter experiments in Appendix G

**Weaknesses:**

- [W1] The authors argue that the -Charge and -Energy tasks  are 'inherently long-range' due to (i) their large size/diameter, 17-40 hops, and (ii) the underlying task; they say:
	- "The three-dimensional configuration of molecules greatly intensifies this task complexity, as distant atoms in the graph topology can still exert significant influence on electronic properties and total energy."
	- [W1.1] This makes sense to me, but do you have a source for it?
	- To put it another way; large diameter/graph size and intuition about the underlying task are strong indicators that a task is long-range, but do not prove it necessarily. Your experiments, in which deeper models/Transformers dominate, would also indicate that long-range interactions are present but this is somewhat circular (i.e., if the dataset is an evaluator of model LR capability, its validity as an LR dataset  cannot be based on model performance).
	- [W1.2] With this in mind, I would like to see a more theoretically motivated justification for the long-rangedness of the real-world datasets. (Echo-Energy is a graph-level task, but this is not necessarily inherently long-range; it could well be simply a sum over local information applied at readout.) For example, you cite Bamberger et al. (2025), who propose a range measure, as evidence for the questionable long-rangedness of LRGB. Why not apply this measure to your benchmark?
- [W2] I think experiments would be improved by the inclusion of more than one graph Transformer, perhaps something more recent than GPS — e.g. GRIT [1], Graph-Mixer [2]

---

[1] Ma, Liheng, et al. "Graph inductive biases in transformers without message passing." International Conference on Machine Learning. PMLR, 2023.

[2] He, Xiaoxin, et al. "A generalization of vit/mlp-mixer to graphs." International conference on machine learning. PMLR, 2023.

**Questions:**

- [Q1] Line 284: What is 'total molecular energy'? Is it simply the sum of individual node energies?
- [Q2] Line 1109: Your hyperparameter search for Drew only goes up to 4 layers, whereas it goes up to 40 for other models. As I understand it, Drew is an MPNN, with a receptive field determined by the number of layers. Does this not affect its LR capability? Why such a small search window?
- [Q3] On the task of predicting atomic partial charges —
	- Is this a problem for which GNNs have already shown promise? Do you have a source for this? It was not obvious to me from the text
	- Can you contextualise the experimental results a little? I.e. do any of the models do a good enough job to reasonably replace/approximate expensive DFT computation?
	- To put it another way, it isn't clear to me whether GNNs are actually reasonable for solving this problem, or whether this is just an instructive task for GNN benchmarking

Misc:
- Line 208: Is this an appropriate use of 'skip connection'? I would tend to associate it with residual connections
- Line 454: I would suggest adding a sentence here about the contents of the experiments in Appendix G, as I think they are quite interesting
- Typos:
	- Line 104: 'on the need ~~of~~ **for** a new benchmark'
	- Line 380: 'differently, graphcon do not'

---

> ### Author Response · Authors · 2025-11-21
> **Response**
>
> We thank the Reviewer for their thorough reading and the very positive assessment of our paper, and we greatly appreciate the recognition of (i) the need for new long-range benchmarks, (ii) the sound motivation and design of the ECHO suite, and (iii) the care and effort reflected in our experimental setup and documentation. We are especially grateful for the encouraging comment that “ECHO could become a new and sorely-needed standard benchmark for the community”. We also thank the Reviewer for noting the clarity of our presentation and figures. Below, we address each comment point-by-point.
>
>
> W1 & Q1)
>
> We thank the Reviewer for their willingness to improve the quality of our work. The chemico-physical evidence that our molecular tasks are long-range is supported by \[1,3,4\], which shows that distant atoms can significantly alter charge distributions and total energy via long-range electrostatics and charge transfer. To give a more practical example, the total molecular energy cannot be computed as the sum of per-atom energies, since it consists of several quantum-mechanical long-range interactions, e.g., electron-nuclear, electron-electron, and nuclear-nuclear contributions at the chosen level of theory \[1\]. Specifically, the total energy, i.e., single-point energy, is obtained by solving the Schrödinger equations for a fixed molecular geometry \[1, 2\]. Moreover, empirical and theoretical evidence \[3,4\] supports that partial atomic charges and molecular energies are influenced by non-local electronic effects. Therefore, this discussion, together with our empirical results (showing that model performance improves with larger numbers of message-passing steps) indicates that both ECHO-Charge and ECHO-Energy inherently require modeling interactions over large hops. We included this discussion in the revised paper in Section 3.2.
>
> W2)
>
> We thank the Reviewer for the suggestion, and we agree that additional Transformer baselines would strengthen the study. We are currently conducting experiments with GRIT; however, the completion of  ECHO-Energy and ECHO-Charge has been delayed due to an unexpected server outage (hence also the slight delay in our rebuttal). In the following table we report the results for the ECHO-Synth task with respect to the best performing models, i.e., GPS and SWAN.
>
> | Model       | Diam ↓| Ecc ↓       | SSSP ↓ |
> |--------|---------------|------------|------------|
> |GPS | 2.160 ± 0.098 | **4.758 ± 0.021** | 0.472 ± 0.050
> |SWAN | 1.121 ± 0.070 | 4.840 ± 0.045 | 0.896 ± 0.232
> |GRIT | **1.014 ± 0.014**  |  5.091 ± 0.158 | **0.121 ±  0.013** |
>
> These results show that GRIT is among the best performing models on the ECHO-Synth. We included these results in the revised manuscript (Table 1), and we are currently running a model selection on the molecular benchmarks that will be included as soon as they are completed.
>
>
> Q2)
>
> The Reviewer is right in noting that  our DRew depth search goes up to 4 layers. However, DRew performs multi-hop aggregation (up to 10 hops per layer), yielding an effective receptive field of  $4 \times 10 = 40$ hops, comparable to the ranges explored by the other architectures. We added this clarification in the paper, (Appendix F, Table 7).
>
> Q3)
>
> We appreciate this insightful question. To clarify, our paper has a dual objective. Primarily, we aimed to establish a rigorous benchmark to expose the limitations of current GNN architectures in modeling long-range propagation, which is a known challenge in both synthetic and real-world scenarios.
>
> We consider GNNs a strong candidate for approximating DFT calculations. As detailed in Section 4, the best-performing models show promising results, approaching the precision thresholds required for DFT computations. While they do not yet fully replace DFT-level performance, this result is significant: it confirms that our benchmark not only successfully stresses long-range propagation, but also shows that GNNs are a fundamentally viable strategy for solving the underlying physical problem.
>
> Misc)
>
> We thank the Reviewer for identifying these typos, we revised them in the paper accordingly.
>
> ---
>
> \[1\] Jensen. “Introduction to Computational Chemistry”. John Wiley & Sons. 2017
>
> \[2\] Szaba, et al. Modern Quantum Chemistry: Introduction to Advanced Electronic Structure Theory. MACMILLIAN., 1982\.
>
> \[3\] Ko, et al. "A fourth-generation high-dimensional neural network potential with accurate electrostatics including non-local charge transfer." Nature communications 12.1 (2021): 398\.
>
> \[4\] Shaidu, et al. "Incorporating long-range electrostatics in neural network potentials via variational charge equilibration from shortsighted ingredients." npj Computational Materials 10.1 (2024): 47\.

---

> > ### Comment · Reviewer_5Ph3 · 2025-11-27
> >
> > I thank the authors for their detailed response, which alleviates several of my concerns. A couple of further points:
> >
> > ---
> >
> > > Our empirical results (showing that model performance improves with larger numbers of message-passing steps) [indicate] that both ECHO-Charge and ECHO-Energy inherently require modeling interactions over large hops
> >
> > I think the authors should be careful with this kind of comment, which, as I pointed out in my initial response, is circular — if your benchmark is designed to *evaluate* long-rangedness, then performance gap alone should not be used as evidence that the benchmark is long-range. I still would have liked to see an evaluation using the range measure introduced by Bamberger et al. (2025); you perform an evaluation on a similar measure in your new Appendix K, but dismiss the results as being only suitable to evaluate trained models, not datasets in isolation. But I would argue that such measures *can* be used to evaluate dataset range, if one considers the correlation between range and model performance. This is the argument used by Bamberger et al. (2025), an argument you implicitly support (lines 112, 943) when you use that paper to cite the questionable long-rangedness of LRGB.
> >
> > I am happy with the well-sourced discussion of the underlying physics.
> >
> > ---
> >
> > > The Reviewer is right in noting that our DRew depth search goes up to 4 layers. However, DRew performs multi-hop aggregation (up to 10 hops per layer)...
> >
> > As I understand it, Drew performs $l$-hop aggregation at the $l$th layer, so a 4-layer model should only go up to 4-hop aggregation in the final layer. A cursory look at your code implementation of Drew seems to suggest that this is what is happening. Perhaps I am misunderstanding?

---

> > > ### Author Response · Authors · 2025-12-01
> > >
> > > Dear Reviewer 5Ph3,
> > >
> > > We thank you once again for their review and positive assessment of our work. It was unfortunate that the reviewing activity was frozen before concluding our discussion.
> > >
> > > We are pleased that the Reviewer is "happy with the well-sourced discussion of the underlying physics" theoretically justifying the long-range nature of the tasks, and we sincerely appreciate their consistent support, maintaining an 'Accept' (8) rating in both the initial and final evaluations.
> > >
> > > _Regarding the Reviewer's additional suggestion to further strengthen our work using the range measure from Bamberger et al. (2025):_
> > >
> > > While we believe that the theoretical and physical evidence already provided is sufficient to guarantee the long-range dependencies of our tasks, we are currently running experiments to compute this metric to provide a quantitative perspective over long-rangedness mediated by the trained models' dynamics. Therefore, we are committed to include the results in the final version of the paper as soon as they are completed.
> > >
> > > _Regarding the additional clarification on DRew:_
> > >
> > > The Reviewer noted that in our code DRew explores up to 4 layers in the hyperparameter search space, inferring that this limits the receptive field to 4 hops. However, as noted in our previous response, the implementation of DRew decouples the number of layers from the propagation range; inside each of the 4 layers, the model performs a multi-hop aggregation over 10 propagation steps. Therefore, the total effective receptive field is 40 hops ($4$ layers $\times 10$ steps), rather than 4.
> > >
> > > We hope this clarifies our position and our commitment to incorporating the Reviewer's feedback.

---

> > > > ### Author Response · Authors · 2025-12-01
> > > > **Update: Experimental Results for GRIT Baseline**
> > > >
> > > > We have successfully completed the requested grid search and evaluation for the GRIT [1] baseline, for the missing benchmarks.
> > > >
> > > > The results on the ECHO-Energy and -Charge (Test MAE ± std) are as follows:
> > > > Model | ECHO-Energy ↓ | ECHO-Charge ↓ (x10⁻³) |
> > > > | :--- | :--- | :--- |
> > > > | GRIT | 25.508 $\pm$ 2.507 | 7.134 $\pm$ 6.090 |
> > > >
> > > > We included these results into the final baselines table in the updated version of the paper to ensure a comprehensive comparison.
> > > >
> > > > ---
> > > > [1] Ma, Liheng, et al. "Graph inductive biases in transformers without message passing." International Conference on Machine Learning. PMLR, 2023.

---

### Official Review · Reviewer_UETf · 2025-10-31

**Soundness:** 2
**Presentation:** 2
**Contribution:** 2
**Rating:** 4
**Confidence:** 3

**Summary:**

This paper introduces a benchmark for evaluating long range information propagation in graph learning networks. The benchmark consists of 5 tasks- 3 synthetic graph-theoretic problems (single-source shortest paths, node eccentricity, graph diameter), and 2 real-world datasets focused on molecular graphs (charge and energy) for predicting atomic partial charges and molecular energies. The task design intends to stress on long range propagation based, for example, on the hops these datasets cover. Experiments are done using multiple GNNs, Graph Transformers and differential equations inspired models and insights are presented for models performing good and bad.

**Strengths:**

- There is a gap of datasets on long rage testing in GNNs and although there exists the benchmarks that this paper discusses they have limitations. So the proposed datasets address this.
- The datasets include both synthetic and real world graphs with the real ones standing a good contribution due to the curation as well as the performance trends.
- Empirical results show that models arguably good for long range propagation such as GPS, A-DGN, SWAN, DRew, etc outperform MPNNs, with particularly strong evidence that depth matters.

**Weaknesses:**

- The primary contribution of this work is the collection of molecular dataset as the similar synthetic dataset is known in the prior literature.
- The strong claim in lines 146-147 about the molecular dataset on long range could be disputed since prior molecular dataset also involve long range affects although they could be synthetic targets.
- The paper lacks close discussion/comparison/adaptation of known insights with 2 works which would make it stronger, particularly on quantifying ECHO datasets' long rangeness. First with [1] which has a strinkingly similar multi-task synthetic dataset. Second, with [2] which quantifies long rangeness of a graph dataset.

[1] Corso, Gabriele, Luca Cavalleri, Dominique Beaini, Pietro Liò, and Petar Veličković. "Principal neighbourhood aggregation for graph nets." Advances in neural information processing systems 33 (2020): 13260-13271.
[2] Liang, Huidong, Haitz Sáez de Ocáriz Borde, Baskaran Sripathmanathan, Michael Bronstein, and Xiaowen Dong. "Towards Quantifying Long-Range Interactions in Graph Machine Learning: a Large Graph Dataset and a Measurement." arXiv preprint arXiv:2503.09008 (2025).

**Questions:**

na

---

> ### Author Response · Authors · 2025-11-21
> **Response pt. 1**
>
> We thank the Reviewer for their time and constructive assessment of our work. We greatly appreciate the recognition that **our benchmark helps fill the current gap in datasets for testing long-range propagation** in GNNs, and that **the inclusion of our tasks represents a good contribution to the field.** We are also grateful that **the Reviewer acknowledged the empirical trends observed across architectures and that there is strong evidence that depth plays an important role.** Below, we address the Reviewer’s comments point-by-point in our responses. We hope our clarifications and revisions will address your concerns and prompt a reconsideration of your evaluation and score.
>
> W1)
>
> We agree with the Reviewer that an important aspect of our benchmark is the collection of real-world molecular tasks, which are not only essential for evaluating GNNs’ long-range capabilities but also address central challenges in computational chemistry. Indeed, it required \~2 months of parallel DFT computations to generate our datasets, underscoring the practical relevance and effort behind it. At the same time, we believe that even ECHO-Synth is itself a valuable contribution to the graph learning community. It is designed to deliberately introduce topologies with bottlenecks to stress-test long-range capabilities in a controlled environment (as further detailed in W3). To the best of our knowledge, this is something that has not been explored before.
>
> W2)
>
> We thank the Reviewer for this valuable observation. Our claim is grounded in empirical and theoretical chemico-physical evidence showing that partial atomic charges and total molecular energies depend on non-local electronic interactions. For instance, \[1, 2\] demonstrate that distant atoms can significantly influence charge distributions and energy through long-range electrostatics and charge transfer: effects that cannot be captured by local descriptors alone. Accordingly, both ECHO-Charge and ECHO-Energy inherently require propagating information over the entire graphs. We have clarified this aspect and added the required references in the revised paper (Sec. 3.2).

---

> ### Author Response · Authors · 2025-11-21
> **Response pt. 2**
>
> W3)
>
> We thank the Reviewer for the suggestion. We could not include  \[4\] as it was accepted to NeurIPS 2025 concurrently with  the submission of our paper; nonetheless, we are happy to include a discussion of \[4\] and to better position ECHO-Synth with respect to the tasks in \[3\] in the revised version of our manuscript. We also note that ECHO-Energy and ECHO-Charge are completely novel tasks not covered in either \[3\] or \[4\].
>
> While ECHO-Synth draws inspiration from tasks 1, 2, and 5 in \[3\], it departs from them in a key aspect. The benchmark in \[3\] aims to evaluate a method that improves the expressive power of GNNs. With this aim, they propose a benchmark that uses small graphs, mostly sampled from distributions that yield highly connected structures with small diameters and limited long-range dependencies. In contrast, ECHO-Synth is explicitly designed to stress-test long-range capabilities. To this end, it (i) uses larger graphs (≈84 nodes) and (ii) employs topologies deliberately designed to introduce bottlenecks, increasing the graph diameter (up to 40 hops) and requiring substantially more propagation steps for accurate predictions. These design choices significantly increase the long-range difficulty of the tasks, making ECHO-Synth a more rigorous benchmark for evaluating long-range propagation than \[3\].
>
> Regarding \[4\], which is contemporaneous to our work according to the ICLR guidelines, we view ECHO and \[4\] as complementary efforts addressing the long-range propagation problem. While \[4\] focuses on a single synthetic task on large graphs (up to 569k nodes), our ECHO benchmark proposes five tasks (three synthetic and two real-world) that provide a controlled setting to assess propagation capabilities, and are inherently long-range (as previously discussed). We emphasize that we propose a benchmark that already extends beyond current standards, with the goal of providing a practical, accessible benchmark that balances long-range complexity and usability for the broader community, avoiding digital divide concerns while still reflecting real-world scientific challenges.
>
> Finally, \[4\] proposes a metric for evaluating the long-rangedness of a benchmark that, upon close analysis, appears to be inherently model-dependent, as it relies on the model’s dynamics. To align with their evaluation protocol, we tested this metric on ECHO-Synth–eccentricity, which shares the same objective as \[4\], and it is long-range by design. We found that the same model configuration produces substantially different measures of long-rangedness when evaluated with trained versus randomly initialized weights as shown in Figure 14\. This variability suggests that, while the metric can be used to measure the relative long-rangedness between different models when the dataset is fixed, it may not be used, per-se, to measure the absolute long-rangedness of a specific dataset (and to confront it with others).
>
>  Additionally, the metric is defined only for node-level tasks and therefore cannot be computed on graph-level tasks (e.g., ECHO-Energy), limiting its applicability. In conclusion, since \[4\] (understandably, given the recency of the work) is still in an early stage of dissemination, the available repository does not yet offer sufficient clarity to determine whether our reproduction and evaluation faithfully reflect the authors’ intended setup. Nonetheless, we will be interested in running a more comprehensive analysis once additional implementation details are released.
>
> We included this discussion in our paper Section 2 and 3.1.  Thank you.
>
> ---
>
> [1] Ko, Tsz Wai, et al. "A fourth-generation high-dimensional neural network potential with accurate electrostatics including non-local charge transfer." Nature communications 12.1 (2021): 398.
>
> [2] Shaidu, Yusuf, et al. "Incorporating long-range electrostatics in neural network potentials via variational charge equilibration from shortsighted ingredients." npj Computational Materials 10.1 (2024): 47.
>
> [3] Corso, et al. "Principal neighbourhood aggregation for graph nets." Advances in neural information processing systems 33 (2020): 13260-13271.
>
> [4] Liang et al. "Towards Quantifying Long-Range Interactions in Graph Machine Learning: a Large Graph Dataset and a Measurement." arXiv preprint arXiv:2503.09008 (2025).

---

> > ### Author Response · Authors · 2025-12-01
> >
> > Dear Reviewer UETf,
> >
> > We thank the Reviewer once again for their valuable feedback and regret that the reviewing activity was frozen before we could conclude our discussion.
> >
> > While we believe that the theoretical and physical evidence we provided (further strengthened during the rebuttal) already supports the long-range dependencies of our tasks, and although the contemporaneous work of Liang et al. (2025) may not be ideal for measuring long-rangedness _per se_, we have started running experiments using the metrics introduced in [1] to offer a quantitative assessment of long-rangedness as mediated by the trained models’ dynamics. We are committed to including these results in the final version of the paper as soon as they are completed.
> >
> > We hope this clarifies our dedication to addressing the Reviewer’s feedback.
> >
> > [1] Bamberger et al., On Measuring Long-Range Interactions in Graph Neural Networks, ICML 2025, arXiv.2506.05971

---

### Author Response · Authors · 2025-11-21

We thank the Reviewers for their time, constructive feedback, and positive assessment of our work. We have done our best to address all comments below. Furthermore, we are grateful that the Reviewers highlighted several key strengths in our submission, including that our benchmark helps fill the current gap in datasets for testing long-range reasoning and that the inclusion of both synthetic and real-world molecular tasks represents a meaningful and original contribution. The Reviewers also acknowledged the sound motivation and design of the ECHO suite, the rigorous experimental setup and hyperparameter tuning, and the clarity of our presentation and figures. We are especially grateful for the encouraging comment that _ECHO could become a new and sorely-needed standard benchmark for the community_.

We have already updated all the comments and results in the manuscript (highlighted in blue in this version), and we plan to constantly update the manuscript with the remaining results until the rebuttal deadline. We now address each of the reviews individually below.

---

### Author Response · Authors · 2025-11-29
**Summary of Post-Rebuttal Status and Scores**

We thank the organizers for their transparency regarding the data leak. Given the decision to revert reviews and scores to their pre-rebuttal state, we provide this summary for the new Area Chairs.

During the discussion period, we successfully addressed the comments raised by the reviewers. We are grateful that this productive engagement led to two reviewers maintaining their 'Accept' (8) ratings and one reviewer raising their score from 6 to 8, reflecting a strong consensus on the quality of our work.

Unfortunately, Reviewer UETf did not respond to our rebuttal or engage in the discussion, despite our best efforts to address all their concerns. Due to the sudden termination of the commenting period, we were denied the opportunity to further resolve their issues or have a constructive dialogue.

We respectfully ask the new Area Chairs to consider our comprehensive responses to all reviewers, and we stand by the quality of our submission and the improvements made during the rebuttal.

Kind Regards,

The Authors

---

### Meta-Review · Area_Chair_Gbrh · 2025-12-26

**Summary:**

This paper introduces a benchmark for evaluating graph neural networks (GNNs), specifically designed for long-range graph propagation. Existing long-range benchmarks exhibit performance saturation, and partial subtasks are essentially evaluating short-range interactions. To this end, this paper introduces ECHO (Evaluating Communication over long HOps), which comprises synthetic tasks with specifically designed topological bottlenecks, as well as quantum chemistry tasks based on high-precision DFT calculations. By constructing graphs that have diameters of at least 17 hops, ECHO reveals the limitations of traditional message-passing machines, verifies the advantages of global attention mechanisms (e.g., Graph Transformers), and differential-equation-inspired GNNs in overcoming over-smoothing and over-squashing problems. Notably, ECHO provides rigorous evaluation criteria that combine theoretical depth and practical value for developing the next generation of GNNs, which can better capture complex long-range interactions.

Reviewers generally agreed that the paper addresses an important problem: evaluating the capability of GNNs in capturing long-range interaction. ECHO is a well-designed benchmark that helps fill the current gap in evaluating GNNs on long-range reasoning (UETf, 5Ph3, 5FWW, DTw9). In particular, it combines synthetic tasks aimed at stress-testing bottleneck topologies with real molecular tasks of practical scientific interest (5FWW, DTw9). The empirical evaluation was found to be comprehensive and rigorous, clearly revealing the role of model depth for long-range reasoning by comparing multiple architectures (UETf, 5Ph3, 5FWW, DTw9). The presentation of the paper was also highly rated and found to be clear (5Ph3, 5FWW, DTw9).

After careful consideration of the paper, the reviews, the author’s rebuttal, and the subsequent discussion, the final recommendation is Accept. The decision is mainly based on: the well-designed benchmark fills the gap in evaluating the long-range reasoning capabilities of GNNs (UETf, 5Ph3, 5FWW, DTw9); ECHO efficiently combines synthetic tasks targeting topological bottlenecks with molecular tasks with real-world applications (5FWW, DTw9); ECHO’s comprehensive empirical evaluations clearly reveal performance gaps between different architectures (UETf, 5Ph3, 5FWW, DTw9). In addition, the authors effectively address key technical concerns to further enhance the quality of the paper by (i) providing theoretical supports for the long-range physical interaction of the real-world benchmark; (ii) clarifying the implementation details of the baseline model; and (iii) validating the scalability and robustness of the model through additional experiments.

**Reviewer Concerns:**

The reviewers raised several specific technical issues, including the need for theoretical support for the physics of task long-range interactions (addressed by the authors through additional literatures and discussions), clarification of the implementation details of specific baseline models (addressed by the authors), and questions about the dataset's ability to scale up on large-scale graphs (addressed by the authors through the addition of experiments). The authors provide a detailed rebuttal, including new experimental data and explanations, which satisfy reviewers 5Ph3, DTw9, and 5FWW.

In addition to technical clarifications, a discussion about the novelty of the synthesis benchmark (raised by UETf). The authors claimed that the synthesis benchmark aims to provide a more rigorous long-range stress-testing environment than existing work by introducing unique topological bottlenecks. Although this divergence was not explicitly resolved due to the reviewer's absence during the discussion period, the distinctive value of the work was validated by the remaining reviewers.

**Reviewer Scores:**

Reviewer 5Ph3 and DTw9 had made it clear during the rebuttal period that they are satisfied with the authors' response and have maintained their high scores of "Accept" (8), so their scores are expected to remain stable and firm. Reviewer 5FWW raised the score during the discussion period after seeing the additional experiments.

On the other hand, reviewer UETf (initial score 4) was mainly concerned with the novelty of the synthesis benchmark and the long-range nature of the molecular task. Had this reviewer been able to engage in a follow-up conversation, the key evidence provided by the authors in their rebuttal would have been sufficient to overturn their rejection.

---

### Decision · Program_Chairs · 2026-01-26

Accept (Poster)